**EMBO** *reports*

# DLG1 functions upstream of SDCCAG3 and IFT20 to control ciliary targeting of polycystin-2

Csenge K Rezi [1], Mariam G Aslanyan[2], Gaurav D Diwan [3,4], Tao Cheng[5], Mohamed Chamlali [1], Katrin Junger [6], Zeinab Anvarian[1], Esben Lorentzen [7], Kleo B Pauly[1], Yasmin Afshar-Bahadori[1], Eduardo FA Fernandes[8], Feng Qian [9], Sébastien Tosi[10], Søren T Christensen [1], Stine F Pedersen [1], Kristian Strømgaard[8], Robert B Russell [3,4], Jeffrey H Miner[5], Moe R Mahjoub [5], Karsten Boldt[6], Ronald Roepman [2] & Lotte B Pedersen [1]✉

## Abstract

**Polarized vesicular trafficking directs specific receptors and ion channels to cilia, but the underlying mechanisms are poorly understood. Here we describe a role for DLG1, a core component of the Scribble polarity complex, in regulating ciliary protein trafficking in kidney epithelial cells. Conditional knockout of *Dlg1* in mouse kidney causes ciliary elongation and cystogenesis, and cell-based proximity labeling proteomics and fluorescence microscopy show alterations in the ciliary proteome upon loss of DLG1. Specifically, the retromer-associated protein SDCCAG3, IFT20, and polycystin-2 (PC2) are reduced in the cilia of DLG1-deficient cells compared to control cells. This phenotype is recapitulated in vivo and rescuable by re-expression of wild-type DLG1, but not a Congenital Anomalies of the Kidney and Urinary Tract (CAKUT)-associated DLG1 variant, p.T489R. Finally, biochemical approaches and Alpha Fold modelling suggest that SDCCAG3 and IFT20 form a complex that associates, at least indirectly, with DLG1. Our work identifies a key role for DLG1 in regulating ciliary protein composition and suggests that ciliary dysfunction of the p.T489R DLG1 variant may contribute to CAKUT.**

**Keywords** DLG1; Primary Cilia; SDCCAG3; IFT20; Polycystin-2
**Subject Categories** Cell Adhesion, Polarity & Cytoskeleton; Membranes & Trafficking; Molecular Biology of Disease

## Introduction

Primary cilia are microtubule-based sensory organelles that protrude from the surface of many different vertebrate cell types, including kidney epithelial cells, and play essential roles in regulating various signaling pathways during embryonic development and adult homeostasis. Mutations in ciliary genes lead to deregulated signaling, in turn causing diseases known as ciliopathies. While dysfunctional cilia affect most organs in the body, renal involvement is a key feature of many ciliopathies (Mill et al, 2023). For example, autosomal dominant polycystic kidney disease (ADPKD), one of the most common human monogenic diseases affecting ca. 1:1000 live births, can be caused by mutations in *PKD1* or *PKD2* encoding the cilium-localized transmembrane proteins polycystin-1 (PC1) and polycystin-2 (PC2), respectively, which form a heterodimeric calcium-permeable nonselective cation channel complex essential for tubular differentiation, polarity and diameter in the kidney (Cantero and Cantiello, 2022; Ma et al, 2017; Pazour et al, 2002; Yoder et al, 2002).

Cilia are compartmentalized organelles that are thought to be devoid of protein synthesis machinery, and appropriate trafficking of PC1 and PC2, as well as other ciliary signaling receptors, ion channels and transporters, from their site of synthesis in the ER/Golgi to the ciliary compartment is essential for ciliary biogenesis and function. For example, a mutation that specifically impairs ciliary localization of PC2, one of the best-studied ciliary ion channels, causes PKD in mice (Walker et al, 2019). Additional studies have addressed the molecular mechanisms by which PC2 and other ciliary transmembrane proteins, such as G-protein coupled receptors (GPCRs), are sorted and transported from their site of synthesis in the ER/Golgi to the cilium. These studies have revealed a remarkable complexity and diversity in the mechanisms by which different transmembrane proteins are targeted and transported to the primary cilium (Hu and Harris, 2020; Monis et al, 2017; Nachury, 2018). In the case of PC2, studies of its

[1]Department of Biology, University of Copenhagen, Copenhagen, Denmark. [2]Department of Human Genetics, Research Institute for Medical Innovation, Radboud University Medical Center, Nijmegen, The Netherlands. [3]BioQuant, Heidelberg University, Heidelberg, Germany. [4]Biochemistry Center (BZH), Heidelberg University, Heidelberg, Germany. [5]Department of Medicine (Nephrology Division) and Department of Cell Biology and Physiology, Washington University, St Louis, MO, USA. [6]Institute for Ophthalmic Research, Eberhard Karl University of Tübingen, Tübingen, Germany. [7]Department of Molecular Biology and Genetics - Protein Science, Aarhus University, Aarhus, Denmark. [8]Center for Biopharmaceuticals, Department of Drug Design and Pharmacology, University of Copenhagen, Copenhagen, Denmark. [9]Division of Nephrology, Department of Medicine, University of Maryland School of Medicine, Baltimore, MD, USA. [10]Danish BioImaging Infrastructure Image Analysis Core Facility (DBI-INFRA IACF), University of Copenhagen, Copenhagen, Denmark. ✉E-mail: lbpedersen@bio.ku.dk

 

glycosylation pattern indicated that PC2-containing vesicles destined for the cilium are initially released from the cis-Golgi compartment instead of the trans-Golgi network (TGN) (Hoffmeister et al, 2011; Stoops and Caplan, 2014) although this was questioned by others (Kim et al, 2014a). Regardless, it is believed that shortly after synthesis, PC2 interacts with the ciliary intraflagellar transport (IFT)-B complex subunit IFT20, which is anchored to the cis-Golgi compartment by golgin protein GMAP210/TRIP11, and facilitates the transport of PC2 to the base of the primary cilium (Follit et al, 2008; Follit et al, 2006). Following sorting at the cis-Golgi or TGN, PC2-containing vesicles are thought to be directed toward the plasma membrane or recycling endosomes before PC2 is delivered at the ciliary base where it docks at the transition fibers prior to being imported into the cilium (Hu and Harris, 2020; Monis et al, 2017; Walker et al, 2019). Consistent with recycling endosomes playing a critical role in conferring PC2 targeting to the primary cilium, disruption of recycling endosome-associated proteins, such as retromer-associated proteins and the biogenesis of lysosome-related organelles complex 1 (BLOC-1) complex, or the Rab family small GTPases RAB8 and RAB11, reduces ciliary PC2 levels (Hoffmeister et al, 2011; Monis et al, 2017; Tilley et al, 2018; Yu et al, 2016). Whether and how components located at the plasma membrane contribute to ciliary PC2 trafficking is largely unknown.

Discs large MAGUK scaffold protein 1 (DLG1) is a scaffold protein that belongs to the membrane-associated guanylate kinase homolog (MAGUK) family and is composed of a LIN-2,-7 (L27) domain, three postsynaptic density-95/discs large/zona occludens-1 (PDZ) domains, a SRC homology 3 (SH3) domain and a catalytically inactive guanylate kinase (GUK) domain. First described in *Drosophila*, this evolutionarily conserved scaffold protein is well-known for its role in apical-basal polarity establishment and maintenance in epithelial cells, where it forms a complex with SCRIB and LGL at the basolateral membrane below the adherens junctions (Funke et al, 2005). Consistent with its domain structure, the DLG1 interaction network is vast, and its function extends beyond epithelial cell polarity establishment. For instance, in neurons DLG1 localizes to both the presynaptic and postsynaptic membranes and controls localization and clustering of glutamate receptors and potassium channels by mediating interaction between receptors and intracellular proteins (Gardoni et al, 2007; Lickert and Van Campenhout, 2012; Mauceri et al, 2007; Sans et al, 2001; Tiffany et al, 2000). Several studies also suggested that DLG1 localizes to the cilium–centrosome axis. Specifically, in HT1299 cells DLG1 was reported to localize to mitotic centrosomes in a PTEN-NEK6-Eg5-dependent manner (van Ree et al, 2016), whereas three independent studies found DLG1 in the ciliary proteome of cultured mouse kidney inner medullary collecting duct 3 (IMCD3) cells (Kohli et al, 2017; Mick et al, 2015) and photoreceptor outer segments (Datta et al, 2015), which are modified primary cilia. DLG1 also binds directly to kinesin-3 motor KIF13B (Hanada et al, 2000), which was shown previously to localize dynamically to the primary cilium where it regulated ciliary composition and signaling (Juhl et al, 2023; Schou et al, 2017). However, cilium-associated functions for DLG1 have so far not been reported.

The physiological importance of DLG1 in vertebrates is highlighted by the fact that *Dlg1* gene-trap mutant mice display neonatal lethality, growth retardation, craniofacial abnormalities, and small kidneys characterized by impaired ureteric bud branching and reduced nephron formation (Caruana and Bernstein, 2001; Naim et al, 2005). Similarly, complete knockout of *Dlg1* in the mouse was shown to cause neonatal lethality due to severe defects in e.g. cardiovascular and craniofacial development as well as defective formation of urogenital organs (Iizuka-Kogo et al, 2007; Mahoney et al, 2006). With respect to the latter, *Dlg1* knockout animals appeared to lack the stromal cells that normally lie between the urothelial and smooth muscle layers, and the circular smooth muscle cells in the ureteric smooth muscle were misaligned, giving rise to impaired ureteric peristalsis and hydronephrosis (Mahoney et al, 2006). Conditional ablation of *Dlg1* in the metanephric mesenchyme, which gives rise to the various segments of the mature nephron, resulted in the formation of glomerular cysts, dilated proximal tubules, protein casts, and diffuse areas of inflammation. The kidneys appeared grossly cystic in contrast to controls, and the mice showed elevated levels of blood urea nitrogen and serum creatinine, indicative of renal failure (Ahn et al, 2013).

In humans, *DLG1* was identified as a susceptibility gene for congenital anomalies of the kidney and urinary tract (CAKUT) (Westland et al, 2015), and a missense variant in DLG1 (p.T489R) was indeed identified in a patient with CAKUT (Nicolaou et al, 2016). Furthermore, *DLG1* is deleted in the 3q29 microdeletion syndrome that is characterized by mild-to-moderate mental retardation, a long and narrow face, as well as additional phenotypes such as microcephaly, cleft lip and palate, horseshoe kidney, and hypospadia (Willatt et al, 2005). However, while emerging evidence suggests that mutations in ciliary genes can give rise to CAKUT (Gabriel et al, 2018), it is unclear whether some of the kidney phenotypes observed in mice and patients with *DLG1* mutations, are linked to ciliary defects.

Here we investigated a potential role for DLG1 in ciliary biogenesis and function by using a previously described kidney-specific conditional *Dlg1* mouse knockout model (Kim et al, 2014b), as well as cultured kidney epithelial cells. Loss of *Dlg1* in mouse kidney led to ciliary elongation and cortical cyst formation whereas cell-based proximity labeling proteomics and fluorescence microscopy implicated DLG1 in regulating the ciliary protein composition. Specifically, cilia from cells lacking DLG1 contained less SDCCAG3, IFT20 and PC2 than control cells, and re-expression of wild-type DLG1, but not a CAKUT-associated DLG1 missense variant, could rescue this phenotype. Despite its role in regulating ciliary length and composition in kidney epithelial cells, DLG1 was primarily localized to the lateral plasma membrane in these cells. Finally, in agreement with its role in promoting ciliary localization of SDCCAG3 and IFT20, immunoprecipitation assays indicated that DLG1, SDCCAG3 and IFT20 interact with each other, and Alpha Fold modeling furthermore suggested that SDCCAG3 and IFT20 may bind directly to each other. Our work thus identifies a key role for DLG1, located at the lateral plasma membrane of kidney epithelial cells, in mediating ciliary targeting of PC2 and other proteins and supports emerging evidence linking ciliary dysfunction to CAKUT.

## Results

### Kidney-specific ablation of *Dlg1* in mouse causes ciliary elongation

To investigate possible ciliary functions for DLG1, we analyzed kidneys from *Pax3Cre-Dlg1^{F/F}* mice in which *Dlg1* is conditionally knocked out in the majority of kidney cells. Of note, *Pax3Cre* is expressed in the metanephric mesenchyme that differentiates to form the various segments of the mature nephron (glomerulus,

proximal tubule, loop of Henle, and distal tubule). Therefore, *Pax3Cre-Dlg1$^{F/F}$* mice result in loss of *Dlg1* expression in all nephron (but not ureteric bud) epithelial cell derivatives. These mice display a congenital hydronephrosis phenotype (Fig. 1A) similar to that observed in the global *Dlg1$^{-/-}$* mutant mice (Mahoney et al, 2006), as well as tubular dilations that appeared to be pre-cystic (Fig. 1A) (Kim et al, 2014b). The cystic dilations are consistent with prior experiments, where ablation of *Dlg1* in the metanephric mesenchyme (using Six2-Cre) also resulted in the formation of tubular cysts in the kidney cortex (Ahn et al, 2013). The *Pax3Cre* transgene is also active in urogenital mesenchyme, and it was concluded that the lack of DLG1 in these cells results in the observed structural and functional defects in the ureter that cause hydronephrosis (Kim et al, 2014b). Loss of DLG1 resulted in a significant increase in cilia length and acetylated tubulin staining intensity in pre-cystic nephron tubules (Fig. 1B–D; Appendix Fig. S1), indicating that DLG1 plays an essential role in regulating ciliary biogenesis and/or maintenance during kidney development in vivo. Supportively, knockout of *Dlg1* in mouse kidney cortical collecting duct (mCCD) cells (Montesano et al, 2009) did not affect ciliation frequency but led to significant ciliary lengthening when cells were grown on transwell filters, which ensures full cell polarization, a phenotype that was rescued by stable re-expression of mCherry-DLG1 (Fig. 1E–H). In contrast, under standard culture conditions the *Dlg1$^{-/-}$* mCCD cells displayed cilia of normal length (Appendix Fig. S2A), indicating that the ciliary length phenotype manifests itself only when cells are fully polarized. Quantitative RT-PCR analysis showed that in addition to *Dlg1*, mCCD cells also express *Dlg4* and a small amount of *Dlg3*, but the relative expression levels of these mRNAs and of *Dlg2* were not altered in the *Dlg1$^{-/-}$* cells relative to wild-type (WT) cells (Appendix Fig. S2B,C). Thus, the ciliary length phenotype observed in the *Dlg1$^{-/-}$* cells is not caused by altered expression of *Dlg2, 3, or 4*.

## DLG1 localizes to the lateral plasma membrane in polarized kidney epithelial cells

To address how DLG1 might regulate ciliary length we investigated its subcellular localization in transwell filter-grown mCCD cells by immunofluorescence microscopy (IFM) analysis. Under these conditions, endogenous DLG1 localized to the lateral membrane as expected and was not detected at the cilium–centrosome axis (Appendix Fig. S2D). We similarly investigated the subcellular localization of endogenous and tagged versions of DLG1 in IMCD3 cells, but despite intense efforts we only observed DLG1 localization at the plasma membrane and not the primary cilium of these cells. Since previous proteomics analyses detected DLG1 in cilia of IMCD3 cells and mouse photoreceptor outer segments following actin depolymerization or loss of Bardet-Biedl syndrome proteins (Datta et al, 2015; Kohli et al, 2017; Mick et al, 2015), we surmise that DLG1 may localize to cilia under certain conditions and/or is undetectable in cilia by IFM of mCCD and IMCD3 cells for technical reasons. Supportively, eGFP-DLG1 transiently overexpressed in retinal pigment epithelial (RPE1) cells was highly concentrated at the base of and within the cilium (Appendix Fig. S2E), indicating that DLG localizes to the cilium–centrosome axis under some conditions. Nevertheless, taken together our results suggest that DLG1 regulates ciliary length in polarized kidney epithelial cells indirectly, i.e., at the level of the lateral plasma membrane.

## Loss of DLG1 causes altered ciliary protein content in IMCD3 cells

Ciliary length control is complex and regulated by a variety of factors and signaling pathways that modulate the polymerization/depolymerization of the ciliary axoneme or affect ciliary membrane dynamics; changes in ciliary protein composition that affect signaling output can therefore also affect ciliary length (Avasthi and Marshall, 2012; Pedersen et al, 2012). As an example, kidney epithelial cell cilia from patients with ADPKD or from *Pkd1* and *Pkd2* knockout mice were shown to be elongated, possibly due to altered signaling (Liu et al, 2018; Shao et al, 2020). To investigate how DLG1 might affect ciliary protein composition, we used an unbiased cilium-targeted proximity labeling approach (Mick et al, 2015) by taking advantage of previously described IMCD3 cell lines stably expressing a ciliary NPHP3 [residues 1–203]–BioID2 fusion protein (hereafter called cilia-BioID2) or BioID2 alone (hereafter called BioID2) (Aslanyan et al, 2023). We then knocked out *Dlg1* in these lines with the aim of determining how loss of DLG1 affects the ciliary proteome. Western blot analysis confirmed the loss of DLG1 in both the cilia-BioID2 and BioID2 *Dlg1$^{-/-}$* lines (Appendix Fig. S3A). Meanwhile, IFM analysis of serum-starved cells incubated with biotin and stained with an antibody against acetylated α-tubulin (ciliary axoneme marker), and green-fluorescent streptavidin showed prominent ciliary enrichment of biotinylated proteins in both cilia-BioID2 lines, whereas biotinylated proteins were confined to the cell body of the BioID2 lines, as expected (Appendix Fig. S3B). Under these conditions we did not observe any differences between WT and *Dlg1$^{-/-}$* lines with respect to ciliary length (Appendix Fig. S3C) and ciliation frequency (Appendix Fig. S3D), as observed in standard cultures of mCCD cells (Appendix Fig. S2A). Finally, by quantitative RT-PCR we found that IMCD3 cells express similar amounts of *Dlg1* and *Dlg4* (Appendix Fig. S3E,F) and knockout of *Dlg1* did not cause altered expression of *Dlg2, 3 or 4* in these cells (Appendix Fig. S3G,H).

Having validated our WT and *Dlg1$^{-/-}$* cilia-BioID2 and BioID2 lines, we next analyzed the ciliary proteome of these cells by subjecting them to biotin labeling followed by streptavidin pulldown and mass spectrometry. Mass spectrometry analysis resulted in the identification of a total of 2100 proteins across 6 technical replicates per cell line. Our analysis focused solely on proteins that are potentially altered in the primary cilium; therefore, we disregarded the proteins that were significantly altered in the BioID2 control condition. These were further subdivided into three Tiers based on stringency criteria. Tier 1 ($q$ value $\leq 0.05$ and Sign. A $\leq 0.05$) comprised 118 highly significantly altered proteins, from which 84 proteins were depleted from the cilium, whereas 34 proteins were enriched (Fig. 2A). The rest of the proteins were divided into Tier 2 (Sign. A $\leq 0.05$), Tier 3 ($q$ value $\leq 0.05$), and nonsignificant (NS) when a less stringent cut-off was applied (Fig. 2A; Dataset EV1). Using the Tier 1 proteins identified in our dataset, a comprehensive GO term enrichment analysis was performed to pinpoint the functional roles of the proteins regarding DLG1's impact on cilium composition. This analysis focused on the two GO categories: Biological Process (BP) and Cellular Component (CC) (Fig. 2B,C). The BP terms were, in turn, analyzed separately for the depleted and enriched proteins within the cilium (Fig. 2B). For the depleted proteins, the significant BP terms were pertaining to intraciliary transport, cilia

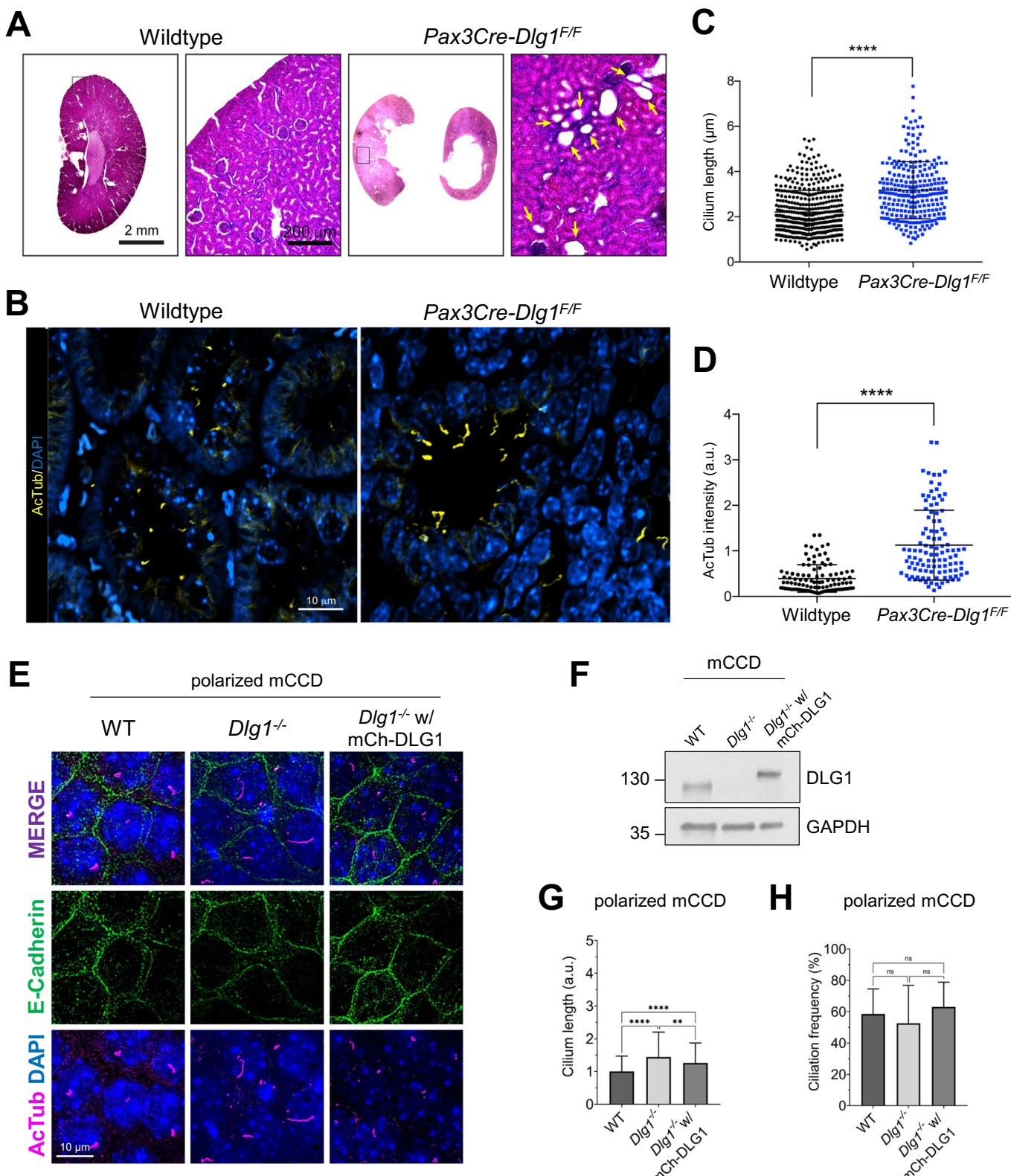

assembly and organization as well several signaling pathways. Moreover, in the GO-CC term category, 15 terms were significant, out of which seven terms were associated with ciliary components (Fig. 2C). On the other hand, for the enriched proteins, BP terms related to the regulation of cell cycle transitions and mitochondrial gene expression were highly significant (Fig. 2B). Altogether, the proximity labeling approach yielded a dataset indicating a role for DLG1 in regulating ciliary composition in IMCD3 cells.

**Figure 1. Loss of *Dlg1* in mouse kidney cells leads to elongated cilia.**

(A) H&E staining of representative kidney sections from wild-type and *Pax3Cre-Dlg1^{F/F}* mice. Black box denotes regions shown as magnified images on the right. Yellow arrows point to cystic tubules in the cortical region. (B–D) Immunofluorescence staining for cilia (acetylated α-tubulin, yellow) and quantification of ciliary length (C) and ciliary acetylated α-tubulin intensity (D) in kidney sections of wild-type (n = 3) and *Pax3Cre-Dlg1^{F/F}* (n = 4) mice at 4 months of age. Data are presented as mean ± SD and statistical analysis was performed using Student's *t*-test. ****P < 0.0001. (E) Representative image of transwell filter-grown mCCD cell lines (mCh-DLG1: mCherry-DLG1). Cilia were visualized using acetylated α-tubulin antibody (AcTub, magenta), cell–cell contacts were visualized with E-cadherin antibody (green) and nuclei were stained with DAPI (blue). (F) Western blot analysis of total cell lysates of the indicated mCCD cells lines using antibodies against DLG1 and GAPDH (loading control). Molecular mass markers are shown in kDa to the left. (G, H) Quantification of ciliary length (G) and frequency (H) in the indicated transwell filter-grown mCCD lines. Ciliary length and ciliation rate were measured using a fully automated MATLAB script (see "Methods" for details). Graphs represent accumulated data from three independent experiments, and statistical analysis was performed using Mann–Whitney *U* test (unpaired, two-tailed). Error bars represent mean ± SD. **P < 0.01; ****P < 0.0001; ns, not statistically significant. Source data are available online for this figure.

## DLG1 is required for ciliary targeting of SDCCAG3 and IFT20 in kidney epithelial cells

To validate the results of our proximity labeling proteomics analysis, we initially focused on the Tier 1 candidates SDCCAG3 and IFT20, which both appeared to be significantly depleted from cilia of *Dlg1^{−/−}* cells compared to WT (Fig. 2A; Dataset EV1). SDCCAG3 is known to bind to the core retromer subunit VPS35 (McGough et al, 2014), and was shown to localize to the base of primary cilia in cultured mammalian cells, including IMCD3 cells, where it also promoted ciliary targeting of PC2 (Yu et al, 2016). Similarly, IFT20 has a well-established role in conferring targeting of PC2 from the Golgi to the primary cilium (Follit et al, 2008; Follit et al, 2006; Monis et al, 2017) and is also part of the IFT-B complex involved in IFT within cilia (Cole et al, 1998). Analysis of ciliated *Dlg1^{−/−}* and WT cilia-BioID2 IMCD3 cells by IFM with antibodies specific for SDCCAG3 confirmed that its ciliary localization is significantly reduced in the *Dlg1^{−/−}* cells compared to WT (Fig. 3A,B), whereas total cellular levels were unchanged (Fig. 3C). Stable expression of mCherry-DLG1 in the *Dlg1^{−/−}* cilia-BioID2 IMCD3 cells could restore ciliary levels of SDCCAG3 to normal (Fig. 3A,B,D), and similar results were obtained in mCCD cells (Fig. 3E–G; see also Fig. 5E,F). Using similar approaches, we confirmed that loss of DLG1 causes reduced ciliary base levels of IFT20 in mCCD cells; in the cilia-BioID2 IMCD3 cells, DLG1 loss had a similar effect although the reduction in IFT20 cilia base staining intensity was not statistically significant compared to WT cells (Appendix Fig. S4A–D). The reason for this cell-type specificity is unclear but may be due to relatively high background staining of IFT20 in the cilia-BioID2 IMCD3 under the specific fixation conditions used. Notably, IFM analysis of kidney sections from WT and *Pax3Cre-Dlg1^{F/F}* mice showed that ciliary levels of SDCCAG3 (Fig. 4A,C) and IFT20 (Fig. 4B,C) are significantly reduced in the *Dlg1* knockout compared to control, indicating that DLG1 promotes ciliary targeting of SDCCAG3 as well as IFT20 in vivo. Since DLG1 was previously shown to interact physically and functionally with exocyst complex component SEC8 (Bolis et al, 2009; Inoue et al, 2006), which in turn mediates ciliary membrane biogenesis and PC2 trafficking (Fogelgren et al, 2011; Monis et al, 2017; Seixas et al, 2016), we also analyzed whether loss of DLG1 affected the ciliary presence of SEC8 in cilia-BioID2 IMCD3 or mCCD cells. However, while this analysis confirmed that SEC8 is concentrated at the ciliary base, we did not observe any significant change in ciliary base levels of SEC8 in *Dlg1^{−/−}* cells compared to WT cells (Appendix Fig. S5). We conclude that DLG1 is required for localizing SDCCAG3 and IFT20, but not SEC8, to the primary cilium of kidney epithelial cells in vitro and in vivo.

## Loss of or acute inhibition or DLG1 impairs ciliary targeting of PC2

Given the known roles of SDCCAG3 and IFT20 in promoting vesicular trafficking of PC2 to the primary cilium (Follit et al, 2008; Follit et al, 2006; Yu et al, 2016), we asked if DLG1 regulates ciliary PC2 trafficking. Although PC2 was not detected in our cilia-BioID2 proximity labeling dataset from IMCD3 cells (Fig. 2A; Dataset EV1), we reasoned this could be due to technical reasons or the cell line used. We therefore used mCCD cells to directly test if inhibition or depletion of DLG1 affected ciliary PC2 levels. First, we cultivated our WT, *Dlg1^{−/−}* and rescue mCCD lines on transwell filters to ensure full polarization of the cells. Confocal 3D imaging showed that the cells were indeed fully polarized under these conditions, and no apparent polarity defects were observed in the *Dlg1^{−/−}* cells compared to the WT and rescue line (Fig. 5A,B). Moreover, the transwell filter-grown *Dlg1^{−/−}* cells also displayed significantly reduced ciliary levels of PC2 compared to the WT cells and this phenotype was rescued by stable expression of mCherry-DLG1 (Fig. 5C,D). For robust and unbiased quantification of ciliary PC2 levels, we employed a MATLAB-based approach (see "Methods" for details) for automatic and high-throughput quantitative analysis of ciliary fluorescent staining intensity in transwell filter-grown mCCD cells. Using this approach, we were also able to confirm our results obtained for SDCCAG3 in mCCD cells grown under standard culture conditions, namely a significantly reduced ciliary presence of SDCCAG3 in *Dlg1^{−/−}* cells compared to WT and rescue lines (Fig. 5E,F).

To confirm that DLG1 regulates ciliary PC2 trafficking, we took advantage of two previously described peptide inhibitors, AVLX-144 (Tat-N-Dimer) and ReTat-N-dimer (Bach et al, 2012) to specifically block the first and second PDZ domain of DLG1 in ciliated WT mCCD cells. We subsequently analyzed the cells by IFM and staining for PC2 in the treated cells; the cilium was visualized by staining with acetylated α-tubulin antibody. We found that treatment of mCCD ciliated cells with both AVLX-144 and ReTat-N-Dimer caused a significant depletion of PC2 from the ciliary base and along the cilium. Importantly, incubation with the control peptide AVLX-144-AA, which is a structurally similar to AVLX-144 and ReTAT-N-dimer, but non-binding to PDZ domains (Bach et al, 2008), did not affect PC2 ciliary levels (Appendix Fig. S6). This result indicates that DLG1 is indeed required for targeting of PC2 to the primary cilium, and that the impaired ciliary targeting of PC2 to the cilium observed upon DLG1 inhibition is not secondary to cytokinesis (Bernabe-Rubio et al,

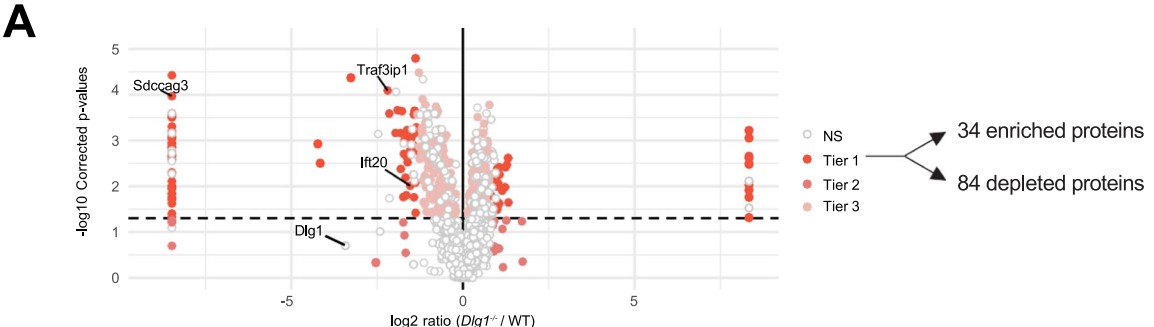

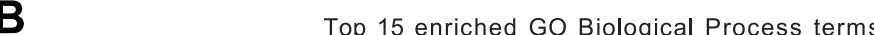

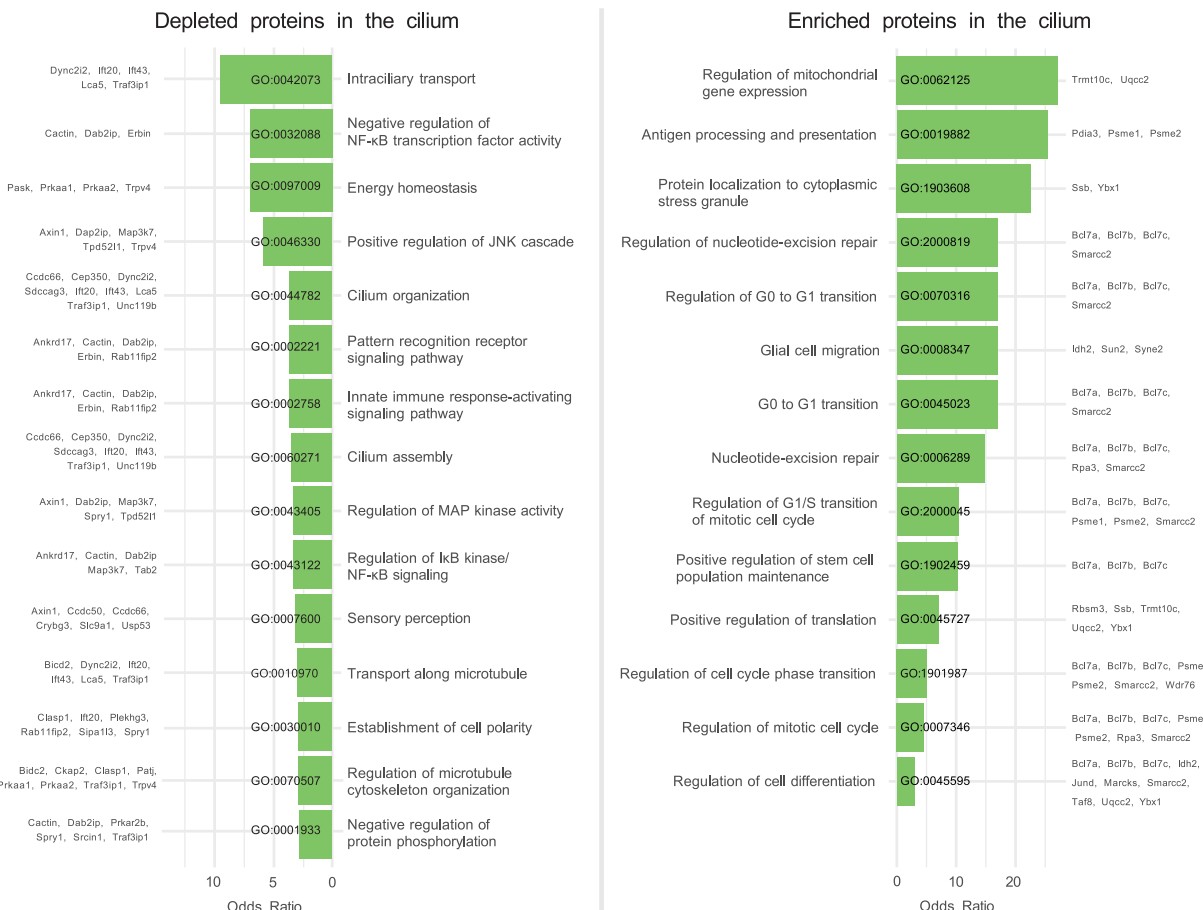

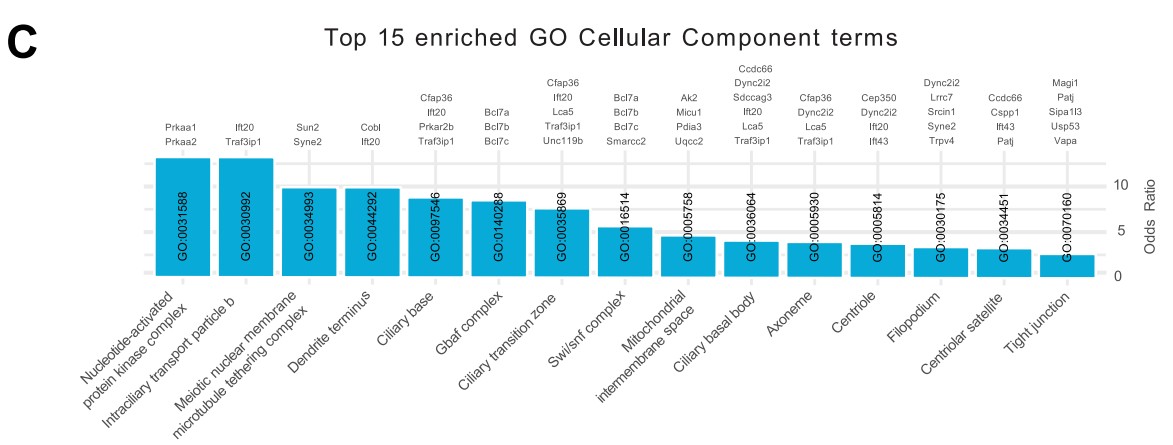

**Figure 2. Analysis of cilia mass spectrometry results.**

(A) Volcano plot visualizing differential protein expression in the ciliary proteome upon disruption of DLG1. Logarithmic ratios are plotted against the negative logarithmic *P*-value of a Student's *t*-test. The proteins are colored according to their significance tier (Tier 1, 2, 3, and non significant (NS)). The total amount of affected candidate ciliary proteins found in Tier 1 upon *Dlg1* depletion are highlighted on the right side of the volcano plot, while the proteins related to this research are marked on the volcano plot. The complete list of identified proteins can be found in Dataset EV1. Note that there are multiple points with the same x axis values on either side of the plot. These proteins were the ones where the median intensity values for either the WT or *Dlg1⁻/⁻* were 0. As a result, the log2 ratio was infinite and we introduced a pseudovalue that is two units above or below the maximum and minimum ratio value across all genes, respectively, to allow plotting of these points. (B, C) Gene Ontology enrichment analysis for biological process (B) and cellular component (C) using the proteins found in Tier 1. The tables show the top 15 terms that are significantly enriched (Fisher's exact test value ≤ 0.05) and are listed in order of their enrichment ratio along with the corresponding GO terms.

2016; Unno et al, 2008) or polarity defects (Lickert and Van Campenhout, 2012).

We conclude that DLG1 is required for targeting PC2 to the primary cilium of kidney epithelial cells, and that the alterations in ciliary composition observed in $Dlg1^{-/-}$ cells are not due to cytokinesis or polarity defects. Furthermore, we conclude that DLG1-mediated ciliary targeting of PC2 requires DLG1's first two PDZ domains.

## A CAKUT-associated DLG1^T507R/T489R missense variant fails to rescue the ciliary phenotype of $Dlg1^{-/-}$ cells

A previous study identified a DLG1 missense mutation (p.T489R; hereafter referred to as DLG1^T489R) in a patient with CAKUT (Table S4 in (Nicolaou et al, 2016)). To investigate a possible ciliary involvement in this disorder, we tested if exogenous expression of the rat equivalent of the human DLG1^T489R missense variant, DLG1^T507R (Fig. 6A), could rescue the ciliary phenotype of $Dlg1^{-/-}$ cells. To this end, we generated a lentiviral construct that we used for stable expression of mCherry-DLG1^T507R in the $Dlg1^{-/-}$ mCCD cells (Fig. 6B). Interestingly, when cells were grown on transwell filters, the DLG1^T507R variant failed to rescue the ciliary elongation phenotype observed in the $Dlg1^{-/-}$ cells (Fig. 6C). Furthermore, while stable expression of mCherry-tagged WT DLG1 fully restored ciliary base levels of IFT20 (Appendix Fig. S4C,D) and SDCCAG3 (Fig. 3E,F) in mCCD cells, this was not the case for the DLG1^T507R variant (Fig. 6D–G). This suggests that ciliary defects may contribute to the CAKUT disease aetiology of patients harboring the DLG1^T489R mutation although we cannot rule out that this mutation separately affects cilia and causes CAKUT.

We also tested the impact of the DLG1^T507R variant on the ciliary levels of PC2 by IFM analysis of transwell filter-grown mCCD cells. Our findings indicated that the DLG1^T507R variant can partially restore the ciliary levels of PC2 to normal in the $Dlg1^{-/-}$ background although not to the same extent as WT DLG1 (Appendix Fig. S7). This suggests that this specific point mutation in DLG1 has a more severe impact on ciliary targeting of SDCCAG3 and IFT20 than PC2, implying that DLG1 promotes ciliary PC2 trafficking not only via SDCCAG3 and IFT20.

## Loss of DLG1 leads to constitutive phosphorylation of TAK1

Upon analysing the GO-BP terms of our proteomics data (Fig. 2B) we noticed that several proteins responsible for regulating MAP kinase activity, such as mitogen-activated protein kinase kinase kinase 7 (MAP3K7, hereafter referred to as Transforming growth factor beta (TGFβ) Activated Kinase 1, TAK1), are diminished in

the primary cilium of $Dlg1^{-/-}$ cells. As TAK1 is linked to the pathogenesis of kidney fibrosis stimulated by TGFβ ligands (Choi et al, 2012; Sureshbabu et al, 2016) and since TGFβ signaling is orchestrated by the primary cilium (Christensen et al, 2017; Clement et al, 2013) we investigated the potential impact of *Dlg1* loss on TGFβ signaling. Upon stimulation with TGFβ-1 ligand, we observed that activation of SMAD2 as evaluated by its phosphorylation on Ser465/467 in the canonical branch of TGFβ signaling was largely unaffected in ciliated $Dlg1^{-/-}$ as compared to WT mCCD cells (Fig. 6H,I). In contrast, we observed that phosphorylation of TAK1 on Thr184/187 and S412 marking full activation of this MAP kinase was increased in unstimulated $Dlg1^{-/-}$ cells as compared to WT cells (Fig. 6J,K). These results indicate that DLG1 takes part in the regulation of sub-pathways in TGFβ signaling, although further studies are needed to delineate the mechanisms by which DLG1 restricts basal levels of TAK1 activation, and whether such mechanisms are controlled at the level of primary cilia.

## DLG1 associates physically with SDCCAG3 and IFT20

Finally, to address the mechanism by which DLG1 promotes targeting of SDCCAG3 and IFT20 to the primary cilium, we tested if DLG1 interacts with these proteins. Indeed, immunoprecipitation (IP) of lysates from HEK293T cells co-expressing GFP-DLG1 and SDCCAG3 or IFT20 fusion proteins indicated that DLG1 interacts with both SDCCAG3 and IFT20 (Fig. 7A,B). Similarly, IP analysis in HEK293T cells demonstrated the interaction between SDCCAG3 and IFT20 (Fig. 7A). To determine the molecular basis for these interactions, we used Alpha Fold modeling (Jumper et al, 2021) and identified a high-confidence interaction between SDCCAG3 and IFT20 (Fig. 7C; Appendix Fig. S8) but did not obtain strong evidence indicative of direct binding of these two proteins to DLG1. Moreover, the predicted interaction between IFT20 and SDCCAG3 is mutually exclusive with binding of IFT20 to its known partner within the IFT-B complex, IFT54 (Taschner et al, 2016) (Fig. 7C). IFT20 was shown previously to interact with the BLOC-1 complex (Monis et al, 2017), and the BLOC-1 complex subunit DTNBP1 (dysbindin) binds directly to DTNA and DTNB (α- and β-dystrobrevin, respectively) of the dystrophin–glycoprotein complex (DGC) (Nazarian et al, 2006). Interestingly, we and others have previously shown that DLG1 as well as its direct interactor, KIF13B, bind to components of the DGC, including UTRN, DTNA and DTNB (Kanai et al, 2014; Morthorst et al, 2022). Furthermore, a high-throughput study indicated that SDCCAG3 also binds DTNBP1 (Huttlin et al, 2021). Therefore, we hypothesize that DLG1 may associate with IFT20 and SDCCAG3 through DTNBP1-DTNA/B interactions but more work is needed to clarify this. In summary, the IP and Alpha Fold modeling results suggest that

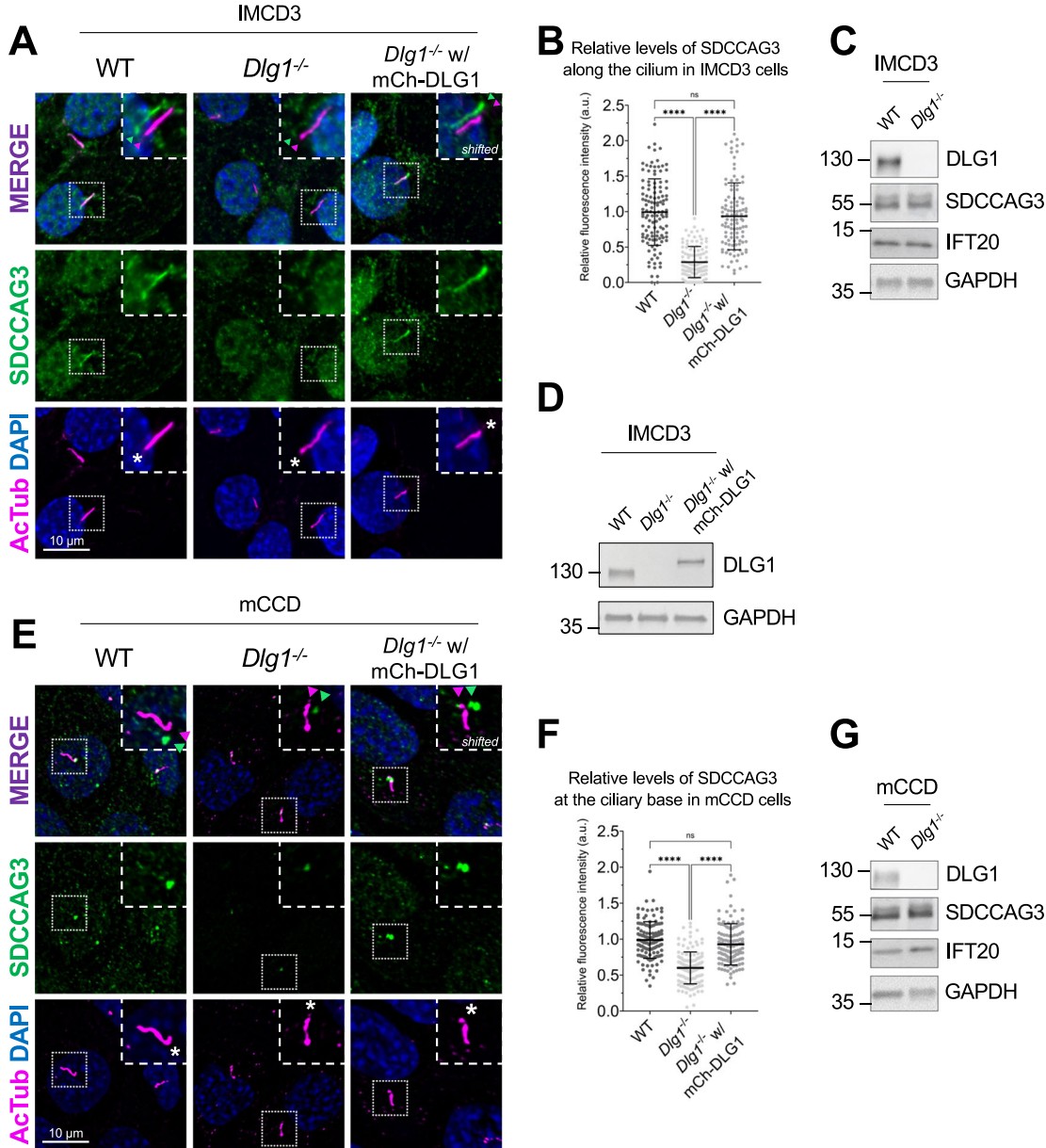

**Figure 3. Loss of DLG1 impairs ciliary localization of SDCCAG3 in IMCD3 and mCCD cells.**

(A) Immunostaining of ciliated cilia-BioID2 IMCD3 cell lines showing comparative SDCCAG3 staining (green) in WT, *Dlg1⁻/⁻* and mCherry-DLG1 (mCh-DLG1) rescue cells. Cilia were stained with antibodies against acetylated α-tubulin (AcTub, magenta), and nuclei visualized with DAPI staining (blue). Insets show enlarged images of cilia, and asterisks mark the ciliary base. The merged insets show primary cilia with channels shifted to aid visualization. (B) Quantification of the relative mean fluorescence intensity (MFI) of SDCCAG3 staining along the cilium of cilia-BioID2 IMCD3 cell lines. Graphs represent WT normalized and accumulated data of three independent biological replicates. The number of dots in each condition represents the total number of primary cilia quantified. Kruskal–Wallis test with Dunn's multiple comparison test was used for the statistical analysis. Error bars represent mean ± SD. ****P < 0.0001; ns, not statistically significant. (C, D) Western blot analysis of total cell lysates of cilia-BioID2 IMCD3 cell lines. Blots were probed with antibodies as indicated, and GAPDH was used as a loading control. Molecular mass markers are shown in kDa to the left. (E) Immunostaining was done with anti-SDCCAG3 antibody (green) and anti-acetylated α-tubulin (AcTub, magenta) in ciliated mCCD WT, *Dlg1⁻/⁻* and mCherry-DLG1 (mCh-DLG1) rescue cells. Nuclei were visualized with DAPI staining (blue). Insets show enlarged images of cilia; asterisks mark the ciliary base. The merged insets show primary cilia with channels shifted to aid visualization. (F) Quantification of the relative MFI of SDCCAG3 staining at the ciliary base of mCCD cell lines. Graphs represent WT normalized and accumulated data of three independent biological replicates. The number of dots in each condition represents the total number of primary cilia quantified. Kruskal–Wallis test with Dunn's multiple comparison test was used for the statistical analysis. Error bars represent mean ± SD. ****P < 0.0001; ns, not statistically significant. (G) Western blot analysis of total cell lysates of mCCD cell lines. Blots were probed with antibodies as indicated, and GAPDH was used as a loading control. Molecular mass markers are shown in kDa to the left. Source data are available online for this figure.

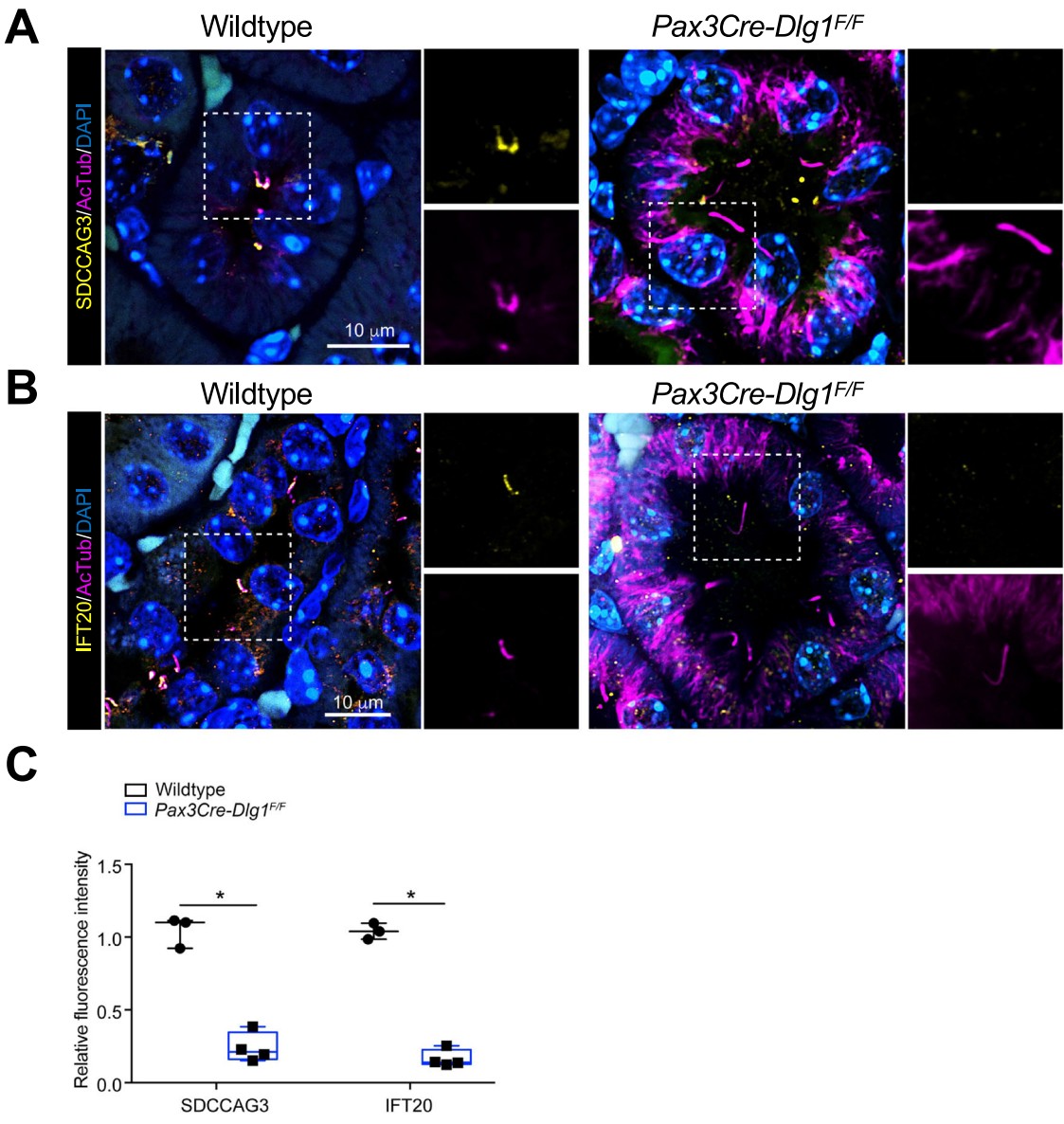

**Figure 4.  Conditional loss of DLG1 in mouse kidney leads to impaired ciliary localization of SDCCAG3 and IFT20.**

(A, B) Immunofluorescence staining of SDCCAG3 (A) or IFT20 (B), both in yellow, and acetylated α-tubulin (AcTub, magenta) in kidney sections from wild-type and *Pax3Cre-Dlg1^F/F* mice. (C) Quantification of relative MFI of SDCCAG3 and IFT20 in cilia of wild-type ($n = 3$) and *Pax3Cre-Dlg1^F/F* ($n = 4$) mice, respectively. The levels from control mice were set to 1, and the ciliary levels from mutant mice were compared to that (i.e., relative fluorescence intensity). Data shown are the average values from each mouse. The vertical segments in box plots show the first quartile, median, and third quartile. The whiskers on both ends represent the maximum and minimum values for each dataset analyzed. Statistical analysis was performed two-tail unpaired *t*-test. *$P < 0.05$, **$P < 0.01$. Source data are available online for this figure.

SDCCAG3 and IFT20 form a heterodimeric complex that associates, at least indirectly, with DLG1.

## Discussion

Here we demonstrated that DLG1 is important for regulating the length and composition of primary cilia in kidney epithelial cells, both in cultured cells and in vivo. Using an unbiased cilium-targeted proteomics approach, we show that loss of DLG1 in IMCD3 cells causes altered ciliary protein content with most of the

affected proteins being diminished in the cilium of *Dlg1^−/−* cells. Specifically, loss DLG1 led to reduced ciliary levels of SDCCAG3 and IFT20, which have both been shown to confer ciliary targeting of PC2 (Follit et al, 2008; Follit et al, 2006; Yu et al, 2016). Consistently, we also established a requirement for DLG1 in promoting ciliary targeting of PC2 in mCCD cells, although our results with a CAKUT-associated missense variant indicated that DLG1 not only confers ciliary targeting of PC2 via SDCCAG3 and IFT20. Reduced ciliary presence of polycystins may at least be partly responsible for the observed ciliary length phenotype of DLG1-deficient cells as loss of PC1 or PC2 was shown to induce

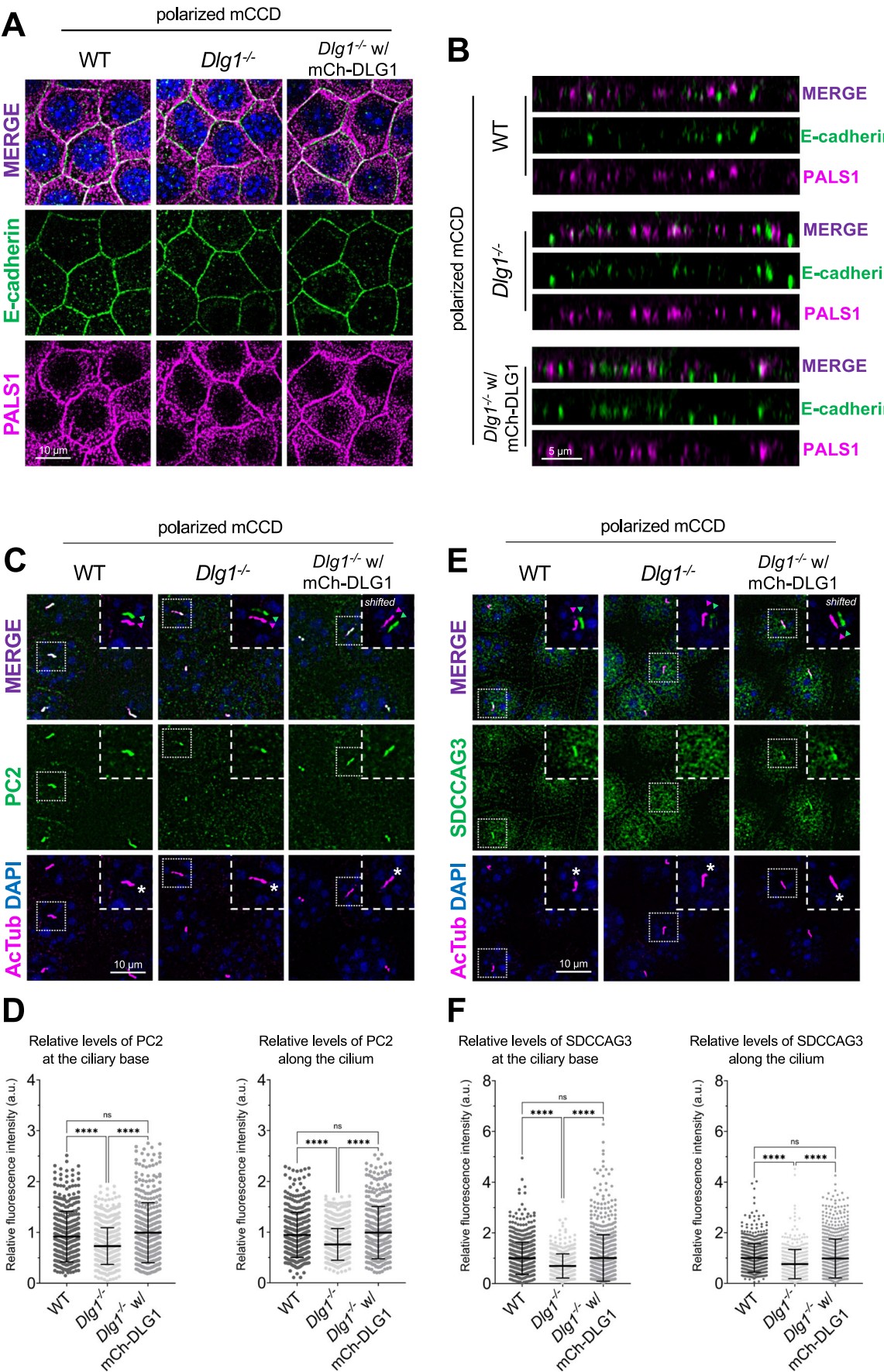

**Figure 5. Loss of DLG1 affects ciliary composition in transwell filter-grown mCCD cells.**

(A, B) Representative top (A) and side view (B) confocal images of transwell filter-grown WT, *Dlg1*$^{-/-}$ and mCherry-DLG1 (mCh-DLG1) rescue lines. The cells were stained for E-cadherin (green) and PALS1 (magenta) to visualize the basolateral membrane and apical-lateral border, respectively. (C, D) IFM analysis of PC2 in the indicated transwell filter-grown cell lines. (C) Immunostaining with anti-PC2 antibody (green) of transwell filter-grown mCCD cell lines, where cilia were visualized with antibodies against α-acetylated tubulin (AcTub, magenta), and nuclei stained with DAPI (blue). Insets show enlarged images of cilia, while the merged insets show primary cilia with channels shifted to aid visualization. Asterisks mark the ciliary base. (D) Quantification of the relative MFI of PC2 along the cilium (right panel) and at the ciliary base (left panel). The MFI of PC2 was measured using the fully automated MATLAB-based quantification. The graphs represent WT normalized and accumulated data of three independent biological replicates. The number of dots in each condition represents the total number of primary cilia quantified. Statistical analysis utilized one-way ANOVA with Tukey's multiple comparison test. Error bars represent mean ± SD. ****$P < 0.0001$; ns, not statistically significant. (E, F) IFM analysis of SDCCAG3 in the indicated transwell filter-grown cell lines. (E) Cilia were visualized with antibodies against acetylated α-tubulin (AcTub, magenta) and nuclei stained with DAPI (blue). Insets show enlarged images of cilia, while the merged insets show primary cilia with channels shifted to aid visualization. Asterisks mark the ciliary base. (F) Quantification of the relative MFI of SDCCAG3 along the cilium (right panel) and at the ciliary base (left panel). The MFI of SDCCAG3 was measured using the fully automated MATLAB-based quantification. The graphs represent WT normalized and accumulated data of three independent biological replicates. The number of dots in each condition represents the total number of primary cilia quantified. Statistical analysis utilized one-way ANOVA with Tukey's multiple comparison test. Error bars represent mean ± SD. ****$P < 0.0001$; ns, not statistically significant. Source data are available online for this figure.

ciliary lengthening in kidney epithelial cells (Liu et al, 2018; Shao et al, 2020), but alternative mechanisms cannot be ruled out. From a human disease perspective PC2 is highly relevant as mutations in its corresponding gene (*PKD2*) cause ADPKD (Mochizuki et al, 1996), and appropriate ciliary localization of PC2 is critical for its function (Hu and Harris, 2020). SDCCAG3 and IFT20 seem to promote ciliary trafficking of PC2 primarily at the level of the recycling endosome and cis-Golgi (Follit et al, 2008; Follit et al, 2006; Monis et al, 2017; Yu et al, 2016), and exocyst complex components also impact the ciliary targeting of PC2 (Monis et al, 2017). The exocyst complex tethers vesicles at target sites before membrane fusion (Heider and Munson, 2012), and DLG1 binds exocyst complex component SEC8 (Bolis et al, 2009; Inoue et al, 2006). However, loss of DLG1 did not affect the ciliary base localization of SEC8 in IMCD3 or mCCD cells.

Although DLG1 may localize to primary cilia under some conditions (Datta et al, 2015; Kohli et al, 2017; Mick et al, 2015), our results indicated that in polarized kidney epithelial cells DLG1 is largely confined to the lateral plasma membrane in agreement with its well-known role as a core component of the Scribble polarity complex. The Scribble complex, which consists of DLG1, Scribble (SCRIB) and lethal giant larvae (LGL), plays a central role in orchestrating epithelial cell polarity (Yamanaka and Ohno, 2008), and Scribble complex components were also implicated in protein cargo sorting and vesicle transport. For example, a study in mouse hippocampal neurons found that DLG1 regulates clathrin-mediated endocytosis of AMPA receptors by recruiting myosin VI and AP-2 adaptor complex to endocytotic vesicles containing these receptors (Osterweil et al, 2005). Furthermore, in *Drosophila* the Scribble complex is required for proper localization of retromer components to endosomes and promotes appropriate sorting of cargo in the retromer pathway (de Vreede et al, 2014), consistent with our finding that DLG1 associates with and regulates ciliary localization of retromer-associated protein SDCCAG3. Studies have demonstrated that deficiency of retromer regulator sorting nexin-17 (SNX17) and SDCCAG3 disrupt ciliogenesis (Wang et al, 2019; Yu et al, 2016). Moreover, the retromer complex interacts with the N-terminal cytoplasmic domain of PC2, and the disruption of this interaction impairs the ciliary localization of PC1 (Tilley et al, 2018). Since DLG1 localizes predominantly to the lateral plasma membrane in polarized kidney epithelial cells our results are consistent with a model whereby DLG1 regulates internalization of ciliary cargoes (SDCCAG3, IFT20, PC2) that are transiently transported to this site prior to their onward journey via recycling

endosomes to the primary cilium (Fig. 7D). Notably, the Na$^+$,HCO$_3^-$ cotransporter NBCn1 (SLC4A7), which localizes at the lateral membrane and primary cilium of polarized kidney epithelial cells, interacts tightly with DLG1 (Severin et al, 2023), and multiple retromer components were identified as putative NBCn1 binding partners in GST pulldown assays (Olesen et al, 2018). Furthermore, our proteomics analysis identified the Na$^+$/H$^+$ exchanger NHE1 (SLC9A1) and the cation channel TRPV4 as Tier 1 candidates depleted from cilia in the *Dlg1*$^{-/-}$ cells (Fig. 2; Dataset EV1). This suggests that DLG1 and the retromer complex may regulate ciliary trafficking of a range of ion channels and transporters, in addition to PC2. Future research should be aimed at addressing this possibility.

Epithelial cells rely on highly organized trafficking machinery to maintain their polarity and carry out their epithelial functions. Such trafficking involves several factors, including sorting signals, cytoskeletal network, vesicle tethering complexes, and Rab and Rho GTPases, that determine the final destinations of each protein (Mellman and Nelson, 2008). Importantly, the cellular microtubule cytoskeleton of polarized epithelial cells is organized very differently compared to mesenchymal cells, with microtubules aligning parallel to the apicobasal axis and extending their plus ends towards the basal surface (Akhmanova and Kapitein, 2022) (Fig. 7D). Therefore, post-Golgi vesicle trafficking in epithelial cells often occurs via indirect transport routes, such as transcytotic or recycling endosomal routes, to ensure delivery of membrane cargo to the apical surface or ciliary compartment (Akhmanova and Kapitein, 2022; Hu and Harris, 2020; Monis et al, 2017). In addition to the lateral plasma membrane functioning as a docking site for ciliary components, prior to their final transport to the cilium, the apical membrane domain may also function as a transit point for ciliary protein trafficking. For example, nephronophthisis proteins NPHP1, NPHP4, and NPHP8 not only localize to the transition zone, but also accumulate at cell junctions, e.g. in polarized kidney epithelial cells (Sang et al, 2011), where they interact with Crumbs polarity complex components (PATJ, PALS1, PAR6) (Delous et al, 2009), which are concentrated at the apical-lateral border, just above the tight junctions (Tan et al, 2020). Conversely, accumulating evidence suggests that components of the Crumbs complex localize to cilia and regulate ciliary assembly or function (Bazellieres et al, 2018; Fan et al, 2004; Morthorst et al, 2022). Notably, our proteomics analysis identified PATJ (INADL) as a Tier 1 candidate depleted from cilia in the *Dlg1*$^{-/-}$ cells (Fig. 2; Dataset EV1) and PATJ was shown previously to interact with PC2 (Duning et al, 2010), suggesting that multiple polarity complexes located along the apical-

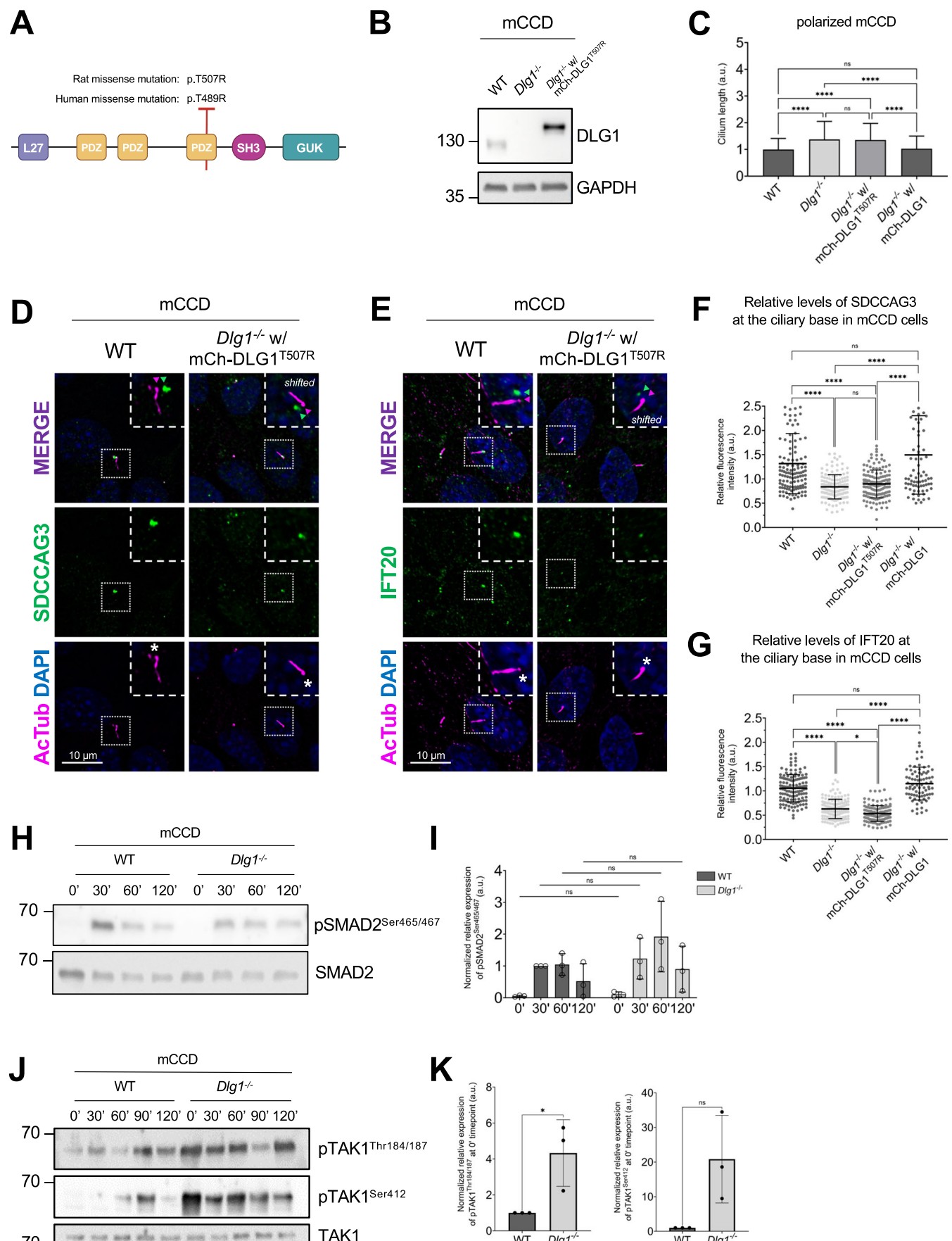

**Figure 6.  A CAKUT-associated DLG1 missense variant fails to rescue ciliary phenotype of _Dlg1$^{-/-}$_ mCCD cells.**

(A) DLG1 protein domain structure and schematic representation and localization of the human CAKUT-associated DLG1$^{T489R}$ variant and the rat counterpart (DLG1$^{T507R}$). The specific human WT DLG1 isoform depicted is DLG1-210 (UniProt Q12959-5), which is encoded by transcript ENST00000422288 (ensembl.org) (Nicolaou et al, 2016); the corresponding WT rat DLG1 isoform is UniProt A0A8I6A5M7. (B) Western blot validation of stable expression of transgenic mutant mCherry-DLG1 (mCh-DLG1$^{T507R}$) in mCCD cells using antibodies as indicated. (C) Ciliary length measurements of indicated cell lines, grown on transwell filters. The ciliary length was measured using the fully automated MATLAB-based quantification. The graphs represent accumulated data of three independent biological replicates. Kruskal–Wallis test with Dunn's multiple comparison test was used for statistical analysis. Error bars represent mean ± SD. ****$P < 0.0001$; ns, not statistically significant. (D, E) IFM analysis of the indicated ciliated cell lines using antibodies against SDCCAG3 (D) or IFT20 (E), both shown in green. Acetylated α-tubulin (AcTub, magenta) was used to stain cilia; nuclei were visualized with DAPI (blue). Insets show enlarged images of cilia, while the merged insets show primary cilia with channels shifted to aid visualization. Asterisks mark the ciliary base. (F, G) Quantification of relative MFI of SDCCAG3 (F) and IFT20 (G) at the ciliary base of indicated mCCD cell lines, based on images as shown in (D, E), respectively. Kruskal–Wallis test with Dunn's multiple comparison test was used for statistical analysis and graphs represent WT normalized and accumulated data of three independent biological replicates, with dots representing total number of cilia analyzed. Error bars represent mean ± SD. ****$P < 0.0001$; ns, not statistically significant. (H) Western blot analysis of total or phosphorylated (p) SMAD2 upon stimulation with TGFβ-1 ligand for indicated times in growth-arrested mCCD cells. (I) Quantifications of protein phosphorylation shown in (H), which represent WT-30′ normalized and accumulated data of three independent biological replicates. Error bars represent mean ± SD. (J) Western blot analysis of total or phosphorylated (p) TAK1 upon stimulation with TGFβ-1 ligand for indicated times in growth-arrested mCCD cells. (K) Quantifications of protein phosphorylation shown in (J), which represent WT-0′ normalized and accumulated data of three independent experimental replicates. Error bars represent mean ± SD. *$P < 0.05$; ns, not statistically significant. Source data are available online for this figure.

basal border of epithelial cells may function together to regulate ciliary protein cargo transport. More studies will be needed to explore this in more detail and define the precise mechanisms involved.

Our cilia proteomics analysis identified several proteins that affect energy homeostasis and NFκB and TGFβ signaling, and which were depleted from cilia of _Dlg1$^{-/-}$_ cells. These include TAK1, whose kinase activity is critical for regulating a variety of cell functions relevant for kidney development and function (Kim and Choi, 2012). Interestingly, our cell-based assays showed that disruption of DLG1 leads to over-activation of TAK1 in line with a recent study, showing that _Dlg1_ deficiency in mouse microglial cells impairs microglial activation and prevents production of inflammatory cytokines (Peng et al, 2021). Furthermore, multiple lines of evidence have shown that alterations in ciliary length and inactivation of polycystins can cause profound metabolic rewiring in the kidney, which likely contributes to the development of PKD (Podrini et al, 2020; Steidl et al, 2023; Walker et al, 2023). Nevertheless, if and how altered ciliary length and composition, as well as dysregulated metabolic, NFκB and TGFβ signaling, contribute to the kidney defects observed in _Dlg1_-deficient mice and human CAKUT patients with _DLG1_ mutations awaits further investigation. However, we note that a more distantly related

DLG1 homolog, DLG5, has been implicated in ciliary biogenesis and function as well as in CAKUT (Chong et al, 2015; Marquez et al, 2021), supporting the involvement of cilia and DLG proteins in this disease.

## Methods

### Mammalian cell culture

IMCD3 cells stably expressing NPHP3[residues 1–203]-BioID2 (hereafter called cilia-BioID2) and BioID2 alone (hereafter called BioID2) have been described previously (Aslanyan et al, 2023). IMCD3 cells were cultured in DMEM/F-12, GlutaMAX Supplement (Gibco, Cat #31331-093) medium supplemented with 10% fetal bovine serum (FBS; Gibco, Cat #10438-026) and 1% Penicillin–Streptomycin (Sigma-Aldrich, Cat #P0781). The immortalized mCCD parental/WT cell line was generously provided by Dr. Eric Féraille (University of Lausanne, Switzerland) and has been described previously (Montesano et al, 2009). The mCCD cells were cultured as described in (Montesano et al, 2009), and RPE1 cells stably expressing SMO-tRFP (Lu et al, 2015) were cultured and transfected as described

**Reagents and tools table**

| Reagent/resource | Reference or source | Identifier or catalog number |
|---|---|---|
| **Experimental models** | | |
| _Pax3Cre_ strain with _Dlg1$^{+/+}$_ alleles; C57BL/6J-CBA/J mixed background (_M. musculus_) | (Kim et al, 2014b) | Wild-type |
| _Pax3Cre_ strain with _Dlg1$^{f/f}$_ alleles; C57BL/6J-CBA/J mixed background (_M. musculus_) | (Kim et al, 2014b) | _Pax3Cre-Dlg1$^{f/f}$_ |
| IMCD3 Flp-In cell line (_M. musculus_) | (Aslanyan et al, 2023) | Wild-type (parental) |
| IMCD3 Flp-In w/ cilia-BioID2 cell line (_M. musculus_) | (Aslanyan et al, 2023) | Wild-type with cilia-BioID2 at Flp-In locus |
| IMCD3 Flp-In w/ BioID2 cell line (_M. musculus_) | (Aslanyan et al, 2023) | Wild-type with BioID2 at Flp-In locus |
| mCCD cell line (_M. musculus_) | (Gaeggeler et al, 2005) | Wild-type (parental) |
| HEK293T cell line (_H. sapiens_) | ATCC | Cat #CRL-3216 |
| IMCD3 Flp-In w/ cilia-BioID2 _Dlg1$^{-/-}$_ cells (_M. musculus_) | This study | Pool of knockout cells |

| Reagent/resource | Reference or source | Identifier or catalog number |
|---|---|---|
| IMCD3 Flp-In w/ BioID2 $Dlg1^{-/-}$ cells (M. musculus) | This study | Pool of knockout cells |
| mCCD $Dlg1^{-/-}$ cell line (M. musculus) | This study | Clone A8 |
| IMCD3 Flp-In w/ cilia-BioID2 $Dlg1^{-/-}$ w/mCherry-DLG1 cells (M. musculus) | This study | Pool/rescue line |
| mCCD $Dlg1^{-/-}$ w/ mCherry-DLG1 cells (M. musculus) | This study | Pool/rescue line |
| mCCD $Dlg1^{-/-}$ w/ mCherry-DLG1$^{T507R}$ cells (M. musculus) | This study | Pool/mutant line |
| hTERT-RPE1 w/SMO-tRFP cell line (H. sapiens) | (Lu et al, 2015) | RPE1 SMO-tRFP |
| DH10B cell line (E. coli) | Lab stock | N/A |
| **Recombinant DNA** | | |
| pSpCas9(BB)-2A-Puro (PX459) V2.0 | Addgene | Cat #62988 |
| gRNA 1/pSpCas9 | This study | pSpCas9-gRNA 1 |
| gRNA 2/pSpCas9 | This study | pSpCas9-gRNA 2 |
| gRNA 3/pSpCas9 | This study | pSpCas9-gRNA 3 |
| gRNA 4/pSpCas9 | This study | pSpCas9-gRNA 4 |
| *Mus musculus* SDCCAG3/ pCMV6-Myc-DDK | Origene | Cat #MR217984 |
| *Homo sapiens* IFT20/ pcDNA5.1- 6xHis-3xFLAG-TEV | Made by Michael Tascher from Esben Lorentzen's lab using standard approaches as in (Taschner et al, 2016) | HFT-IFT20 |
| pEGFP-C1 | Clontech | eGFP |
| *Mus musculus* IFT20/ pEGFP-N1 | (Follit et al, 2006) | IFT20-eGFP |
| *Rattus norvegicus* DLG1/pEGFP-C1 | (Wu et al, 1998) | eGFP-DLG1 |
| pENTR220-mCherry-C1 | (Campeau et al, 2009) | N/A |
| pCDH-EF1a-Gateway-IRES-BLAST | (Campeau et al, 2009) | pCHD |
| pMD2.G | (Goncalves et al, 2021) | N/A |
| pCMVΔ-R8.2 | (Goncalves et al, 2021) | N/A |
| *Rattus norvegicus* DLG1/pENTR220-mCherry-C1 | This study | pENTR220-mCherry-DLG1 |
| *Rattus norvegicus* mCherry-DLG1/ pCDH-EF1a-Gateway-IRES-BLAST | This study | pCDH-mCherry-DLG1 |
| *Rattus norvegicus* mCherry-DLG1$^{T507R}$/ pCDH-EF1a-Gateway-IRES-BLAST | This study | pCDH-mCherry-DLG1$^{T507R}$ |
| **Antibodies and dilutions** | | |
| Anti-alpha-tubulin (mouse monoclonal); WB (1:10,000) | Sigma-Aldrich | Cat #T5168 |
| Anti-acetylated alpha-tubulin (mouse monoclonal); IFM (1:2000), IHC (1:2000) | Sigma-Aldrich | Cat #T7451 |
| Anti-acetylated alpha-tubulin (rabbit monoclonal); IFM (1:2000) | Abcam | Cat #ab179484 |
| Anti-ARL13B (rabbit polyclonal); IFM (1:500) | Proteintech | Cat #17711-1-AP |
| Anti-DLG1 (rabbit polyclonal); IFM (1:750), WB (1:1000) | Abcam | Cat #ab300481 |
| Anti-DLG1 (rabbit polyclonal); WB (1:600) | Thermo Scientific | Cat #PA1-741 |
| Anti-E-Cadherin (rabbit polyclonal); IFM (1:1000) | Cell Signaling Technology | Cat # 3195 |

| Reagent/resource | Reference or source | Identifier or catalog number |
|---|---|---|
| Anti-FLAG (mouse monoclonal); WB (1:1000) | Sigma-Aldrich | Cat #F1804 |
| Anti-GAPDH (rabbit polyclonal); WB (1:1000) | Cell Signaling Technology | Cat #2118 |
| Anti-GFP (chicken polyclonal); WB (1:1000) | Abcam | Cat #ab13970 |
| Anti-GFP (rabbit polyclonal); WB (1:500) | Sigma-Aldrich | Cat #SAB4301138 |
| Anti-IFT20 (rabbit polyclonal); IFM (1:200), IHC (1:100), WB (1:500) | Proteintech | Cat #13615-1-AP |
| Anti-PALS1 (mouse monoclonal); IFM (1:1000) | Santa Cruz Biotechnology | Cat #sc-365411 |
| Anti-PC2 (rabbit polyclonal); IFM (1:1000), WB (1:600) | PKD Research Resource Consortium | N/A |
| Anti-PC2 (mouse monoclonal); IFM (1:500), WB (1:1000) | Santa Cruz Biotechnology | Cat #sc-28331 |
| Anti-SDCCAG3 (rabbit polyclonal); IFM (1:600), IHC (1:100), WB (1:1000) | Proteintech | Cat #15969-1-AP |
| Anti-SMAD2 (rabbit polyclonal); WB (1:200) | Cell Signaling Technology | Cat #5339 |
| Anti-pSMAD2$^{Ser465/467}$ (rabbit polyclonal); WB (1:200) | Cell Signaling Technology | Cat #3108 |
| Anti-rSEC8 (mouse monoclonal); IFM (1:1000), WB (1:2000) | Enzo Life Sciences | Cat #ADI-VAM-SV016 |
| Anti-TAK1 (rabbit polyclonal); WB (1:300) | Cell Signaling Technology | Cat #4505 |
| Anti-pTAK1$^{Ser412}$ (mouse monoclonal); WB (1:200) | Bioss Antibodies | Cat #bs-3435R |
| Anti-pTAK1$^{Thr184/187}$ (mouse monoclonal); WB (1:200) | Bioss Antibodies | Cat #bs-3439R |
| Anti-Mouse-AF488 (donkey polyclonal); IFM (1:600) | Invitrogen | Cat #A-21202 |
| Anti-Mouse-AF568 (donkey polyclonal); IFM (1:600) | Invitrogen | Cat # A-10037 |
| Anti-Rabbit-AF488 (donkey polyclonal); IFM (1:600) | Invitrogen | Cat # A-21206 |
| Anti-Rabbit-AF568 (donkey polyclonal); IFM (1:600) | Invitrogen | Cat #A-10042 |
| Anti-Chicken-HRP (goat polyclonal); WB (1:6000) | Invitrogen | Cat #A-16054 |
| Anti-Mouse-HRP (goat polyclonal); WB (1:10000) | Agilent Technologies, Inc. | Cat #P0447 |
| Anti-Rabbit-HRP (swine polyclonal); WB (1:10,000) | Agilent Technologies, Inc. | Cat #P0399 |
| **Oligonucleotides and other sequence-based reagents** | | |
| *M. musculus Dlg1* exon 5 sgRNA | Eurofins Genomics; this study and (Doench et al, 2016) | sgRNA 1; 5'-TTCTCCACAAGTCACAAATG-3' |
| *M. musculus Dlg1* exon 8 sgRNA | Eurofins Genomics; this study (Doench et al, 2016) | sgRNA 2; 5'-TTGAGTCATCTCCAATGTGT-3' |
| *M. musculus Dlg1* exon 9 sgRNA | Eurofins Genomics; this study (Doench et al, 2016) | sgRNA 3; 5'-TGCGATTGTATGTGAAAAGG-3' |
| *M. musculus Dlg1* exon 14 sgRNA | Eurofins Genomics; this study (Doench et al, 2016) | sgRNA 4; 5'-GGGTCGATATTGCGCAACGA-3' |
| *M. musculus Gapdh* RT-qPCR primers | Eurofins Genomics; this study | sense 5'- TGTCCGTCGTGGATCTGAC-3'; antisense 5'-CCTGCTTCACCACCTTCTTG-3' |

| Reagent/resource | Reference or source | Identifier or catalog number |
|---|---|---|
| *M. musculus 18 S rRNA* RT-qPCR primers | Eurofins Genomics; this study | sense 5'- GCAATTATTCCCCATGAACG-3'; antisense 5'-AGGGCCTCACTAAACCATCC-3' |
| *M. musculus Dlg1* RT-qPCR primers | Eurofins Genomics; this study | sense 5'-CGAAGAACAGTCTGGGCCTT-3'; antisense 5'-GGGGATCTGTGTCAGTGTGG-3' |
| *M. musculus Dlg2* RT-qPCR primers | Eurofins Genomics; this study | sense 5'- TGCCTGGCTGGAGTTTACAG-3'; antisense 5'-TTTTACAATGGGGCCTCCGC-3' |
| *M. musculus Dlg3* RT-qPCR primers | Eurofins Genomics; this study | sense 5'- GAGCCAGTGACACGACAAGA-3'; antisense 5'-GCGGGAACTCAGAGATGAGG-3' |
| *M. musculus Dlg4* RT-qPCR primers | Eurofins Genomics; this study | sense 5'- GGGCCTAAAGGACTTGGCTT-3'; antisense 5'-TGACATCCTCTAGCCCCACA-3' |
| *Rattus norvegicus Dlg1* PCR primer | Eurofins Genomics; this study | rDLG1.kpnl; 5'-CCGGTACCCCGGTCCGGAAGCAAGATAC-3' |
| *Rattus norvegicus Dlg1* PCR primer | Eurofins Genomics; this study | rDLG1.notl; 5'- CCGCGGCCGCTCATAATTTTTCTTTTGCTGGGACCCAG -3' |
| **Chemicals, enzymes, and other reagents** | | |
| Biotin | Sigma-Aldrich | Cat #B4501 |
| Blasticidin S | Gibco | Cat #R21001 |
| DAPI; IFM (1:5000) | Sigma-Aldrich | Cat. #D9542 |
| Dexamethasone | Sigma-Aldrich | Cat # D4902 |
| DLG-specific inhibitor | WuXi ApTtec (Shanghai, China) | AVLX-144 or Tat-N-dimer; described in (Bach et al, 2008; Bach et al, 2012) |
| DLG-specific inhibitor | WuXi ApTtec (Shanghai, China) | ReTat-N-dimer; described in (Bach et al, 2008; Bach et al, 2012) |
| DMEM, high glucose, pyruvate | Gibco | Cat #41966029 |
| DMEM/F-12, GlutaMAX Supplement | Gibco | Cat #31331-093 |
| EGF (epidermal growth factor) | Sigma-Aldrich | Cat #SRP3196 |
| Epredia Immu-Mount | Epredia | Cat #9990402 |
| FastDigest BbsI (BpiI) | Thermo Scientific | Cat #FD1014 |
| FastDigest KpnI | Thermo Scientific | Cat #FD0524 |
| FastDigest NotI | Thermo Scientific | Cat #FD0593 |
| Fetal Bovine Serum (FBS) | Gibco | Cat #31331-093 |
| Gateway LR Clonase II Enzyme mix | Invitrogen | Cat #11791020 |
| Holo-transferrin | Sigma-Aldrich | Cat #T0665 |
| Insulin | Sigma-Aldrich | Cat #I6634 |
| Lipofectamine 3000 Transfection Reagent | Invitrogen | Cat #L3000015 |
| non-PDZ-binding control inhibitor | WuXi AppTec (Shanghai, China) | AVLX-144-AA; (Bach et al, 2008; Bach et al, 2012) |
| NucleoSpin RNA II kit | Macherey-Nagel | Cat #740955.50 |
| Paraformaldehyde | Sigma-Aldrich | Cat #47608 |
| Penicillin–Streptomycin | Sigma-Aldrich | Cat #P0781 |
| Puromycin | Invitrogen | Cat #A11138-03 |
| Sodium selenite | Sigma-Aldrich | Cat #S5261 |
| Streptavidin, Alexa Fluor 488 Conjugate; IFM (1:1000) | Invitrogen | Cat. #S32354 |
| Superscript III Reverse Transcriptase | Invitrogen | Cat #18080-044 |
| SYBR Green | Applied Biosystems | Cat #4309155 |
| T3 (3,3´,5-triiodo-L-thyronine sodium salt) | Sigma-Aldrich | Cat #T6397 |
| T4 DNA Ligase, LC (1 U/µL) | Thermo Scientific | Cat #EL0016 |
| T4 Polynucleotide Kinase (PNK) | New England Biolabs | Cat #M0201S |
| TGF-β1 | R&D Systems | Cat #240-B |

| Reagent/resource | Reference or source | Identifier or catalog number |
|---|---|---|
| **Software** | | |
| Adobe Illustrator 2023 | Adobe | |
| Adobe Photoshop 2023 | Adobe | |
| Alphafold v2.1.0 | (Evans et al, 2022; Jumper et al, 2021) | |
| cellSens 1.18 | Olympus Life Science | |
| Elements AR 5.21 | Nikon | |
| Fiji | (Schindelin et al, 2012) | |
| GraphPad Prism 10.0.1. | GraphPad | |
| Perseus | (Tyanova et al, 2016) | |
| PyMOL v. 2.5 | Schrodinger LLC, https://pymol.org | |
| topGO package in R | (Alexa and Rahnenfuhrer, 2023) | |
| **Other** | | |
| Nunc™ Polycarbonate Membrane Inserts in Multidishes | Thermo Scientific | Cat #140652 |

in (Juhl et al, 2023). Human embryonic kidney (HEK) 293 T cells were from ATCC (Cat #CRL-3216) and were cultured in high-glucose DMEM (Gibco, Cat #41966-052) supplemented with 10% FBS and 1% penicillin–streptomycin.

All cell lines were grown in a 95% humidified incubator at 37 °C with 5% $CO_2$. To induce ciliogenesis, IMCD3 cells were grown in plain DMEM/F-12, GlutaMAX Supplement for 24 h, while mCCD cells were grown in starvation medium, where the serum and hormone-deprived DMEM/F-12, GlutaMAX Supplement medium was supplemented with 5 μg/ml holo-transferrin (Sigma-Aldrich, Cat # T0665) and 60 nM sodium selenite (Sigma-Aldrich, Cat #S5261) for 24 h. All cell lines were routinely tested for *Mycoplasma* contamination by standard approaches.

### Transwell culture system

For setting up fully polarized epithelial cells, mCCD cells were grown in full DMEM/F-12, GlutaMAX Supplement medium as described above, using Thermo Scientific™ Nunc™ Polycarbonate Membrane Inserts in Multidishes (Thermo Scientific, Cat #140652), which have a pore size of 0.4 μm. This was done for a duration of 10 days before proceeding with further experiments. The medium was replaced every 3 to 4 days. For IFM analysis, the polarized mCCD cells were fixed and membrane inserts were excised and treated as described in the general IFM protocol (see below).

### Generation of *Dlg1*⁻/⁻ cell lines

To knock out *Dlg1* in the kidney epithelial cell lines, we employed CRISPR/Cas9 technology and used four sgRNA sequences from the mouse CRISPR "Brie" Knockout Library (Doench et al, 2016). The sequences are provided in the Reagents and Tools table. The sgRNA spacers were cloned into pSpCas9(BB)-2A-Puro (PX459) V2.0 plasmid (Addgene, Cat #62988) as described previously (Ran et al, 2013). This involved phosphorylating and annealing the two complementary sgRNA oligos, which were then ligated into the BbsI-digested backbone. Then the selected clones were sequenced to

verify the spacer sequence. The parental (WT) cilia-BioID2 and BioID2 IMCD3 lines, and the WT mCCD cells were transfected with the Cas9-gRNA plasmids (pool of all four gRNAs) using reverse transfection with Lipofectamine 3000 Transfection Reagent (Invitrogen, Cat #L3000015) according to the manufacturer's instructions. A day after transfection, cells were treated with 2 μg/ml puromycin (Invitrogen, Cat #A11138-03) for 72 h and then tested for DLG1 protein depletion by western blot analysis. Subsequently, the cells underwent single-cell sorting at the FACS Facility at the Biotech Research & Innovation Centre (University of Copenhagen, Copenhagen, DK). The selected clones were validated by western blot analysis and Sanger sequencing to confirm the occurrence of the indel event.

### Generation of transgenic cell lines

A plasmid containing the full-length rat *Dlg1* coding sequence (Wu et al, 1998) was used as template for cloning the rat *Dlg1* coding sequence into Gateway entry plasmid pENTR220-mCherry-C1 using standard cloning techniques. This entry plasmid was then recombined with pCDH-EF1a-Gateway-IRES-BLAST destination plasmid through LR reaction using the Gateway LR Clonase II Enzyme mix (Invitrogen, Cat #11791020). The cloning vectors used were generously provided by Dr. Kay Oliver Schink (Oslo University Hospital, Norway), and were described in (Campeau et al, 2009). The lentiviral expression plasmids were later subjected to site-directed mutagenesis, performed by GenScript, to create a double-point mutation on the following sites: c.1520 C > G and c.1521 T > A; p.T507R. Lentiviral particles were generated by co-transfecting the lentiviral expression plasmids with second-generation lentiviral packaging vectors pMD2.G and pCMVΔ-R8.2 into HEK293T cells (kindly provided by Dr. Carlo Rivolta, Institute of Molecular and Clinical Ophthalmology Basel, Switzerland) using Lipofectamine 3000 Transfection Reagent (Invitrogen, Cat #L3000015) according to the manufacturer's instructions. The harvested culture medium containing lentiviral particles coding for either WT DLG1 or DLG1^T507R fusion proteins were used to transduce the kidney epithelial cells. Cells were selected using 5–15 μg/ml Blasticidin S (Gibco, Cat

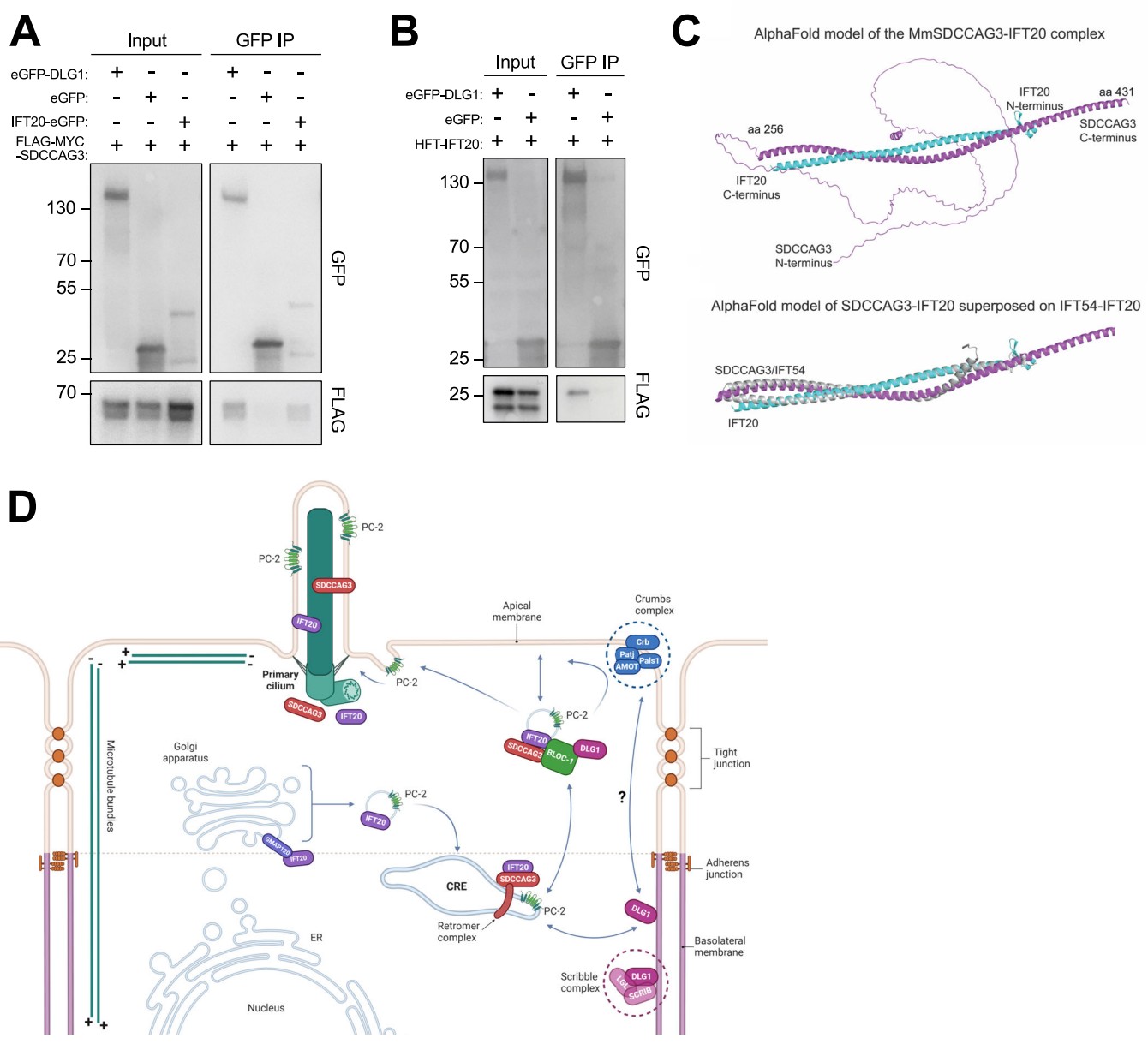

**Figure 7. Analysis of DLG1, IFT20, and SDCCAG3 interactions.**

(A, B) Immunoprecipitation with anti-GFP beads was performed in HEK293T cells transiently expressing FLAG-Myc-SDCCAG3 (A) or FLAG-IFT20 (HTF-IFT20) (B) together with the indicated GFP-fusions. Input and pellet fractions were subjected to SDS-PAGE and western blot analysis using antibodies against FLAG or GFP, as indicated, and GFP expressed alone was used as a negative control. Molecular mass markers are indicated in kDa to the left. (C) Structural prediction for the complex between MmSDCCAG3 (yellow) and MmIFT20 (cyan) in cartoon representation (upper panel). The structure is predicted to be an anti-parallel hetero-dimer coiled coil. The lower panel includes IFT54 showing its binding to IFT20 is mutually exclusive with binding of SDCCAG3 to IFT20. (D) Proposed model for how DLG1 promotes ciliary trafficking of SDCCAG3, IFT20 and PC2. Based on (Hu and Harris, 2020; Monis et al, 2017) and data presented in the current study. CRE common recycling endosome. Source data are available online for this figure.

#R21001) and expression was confirmed with western blotting and live-cell fluorescence microscopy.

## BioID2 proximity labeling

We conducted a proximity labeling experiment, which involved the WT and *Dlg1*$^{-/-}$ cilia-BioID2 lines described above, with the WT and *Dlg1*$^{-/-}$ BioID2 lines as negative controls. Six technical

replicates per cell line were performed. The cells were plated in 15-cm dishes and cultured in normal medium containing DMEM/F-12, GlutaMAX Supplement (Gibco, Cat #31331-093) supplemented as described above. Once the cells had reached 80% confluency, they were stimulated for ciliogenesis for 24 h with the medium described above. Proximity labeling was induced overnight by supplementing the medium with 10 µM Biotin (Sigma-Aldrich, Cat #B4501). The cells were lysed, and samples were prepared for mass

spectroscopy according to a previously published BioID2-based proximity labeling protocol (Aslanyan et al, 2023).

## Mass spectroscopy and data analysis

The samples were analyzed and proteins were identified according to the method described in (Aslanyan et al, 2023). For proteomics data analysis, we used a custom in-house R script that replicates the analysis using the Perseus software (Tyanova et al, 2016). The LFQ intensity values were compared for cilia-BioID2 WT samples versus those for cilia-BioID2 $Dlg1^{-/-}$ samples and for BioID2 WT samples versus BioID2 $Dlg1^{-/-}$ samples. For samples where LFQ intensity values were zero in less than half of the replicates, while having nonzero LFQ intensity values in the other replicates, imputed values were applied drawn from a normal distribution that had a mean that was 1.8 times below the mean of the nonzero values and a standard deviation that was 0.5 times the mean. Subsequently, Student's $t$-test was used for statistical comparisons between the LFQ intensity values of samples as well as the significance A test to infer samples with outlier log2 ratios (high or low). After removing the proteins that were significantly altered in the BioID2 comparison, we devised a three-tier system to classify significant proteins from the cilia-BioID2 comparison. Tier 1 proteins were ones where the corrected $P$-values (Benjamini–Hochberg correction) from the $t$-test were <0.05 as well as significance A test $P$-values were <0.05. Tier 2 proteins included proteins that only had significance A test $P$-values < 0.05 and Tier 3 proteins were the ones that only had corrected $P$-values from the $t$-test <0.05.

## GO term enrichment analysis

To conduct the analysis, the topGO package (Alexa and Rahnenfuhrer, 2023) in R was utilized on the Tier 1 proteins, comprising 118 proteins in total. The approach involved using the GO terms (Biological Process—BP and Cellular Component—CC) linked with all the proteins in the proteomics data analysis and carried out an enrichment analysis for each GO category using Fisher's exact test. Next, a maximum of the top 30 terms were sorted by the Odds ratio and with Fisher's test corrected $P$-value < 0.05 and removed the redundancy in the enriched terms to leave only the terms that were specific and perhaps more informative. This was achieved by removing the other terms that were ancestral in the same GO lineage as the term of interest.

## Immunofluorescence microscopy analysis and live-cell imaging

IMCD3 and mCCD cells were trypsinized (2× concentration, Sigma-Aldrich, Cat #T4174), seeded, and grown on 12-mm diameter glass coverslips. Upon reaching 80% confluence, cells were starved for 24 h to induce robust ciliogenesis using the aforementioned media. The coverslips were fixed in 4% paraformaldehyde (PFA; Sigma-Aldrich, Cat #47608) in PBS for 12 min either at room temperature or at 4 °C, washed with PBS, and incubated in permeabilization buffer (0.2% Triton X-100, 1% BSA in PBS) for 12 min before blocking and antibody incubation. The fixed cells were blocked in 2% (w/v) BSA-based blocking buffer, then incubated with primary antibodies diluted in 2% BSA for 1.5–2 h at room temperature or overnight at 4 °C. After extensive

washing with PBS, cells were then incubated with secondary antibodies diluted in 2% BSA in PBS for 1 h at room temperature. Last, nuclei were labeled with DAPI (Sigma-Aldrich, Cat #D9542). Antibodies and dilutions used in this study for IFM are listed in the Reagents and Tools table. For IFT20 staining, we followed an IFM protocol method described in (Hua and Ferland, 2017) where we briefly washed the cells with cytoskeletal buffer, then immediately fixed them with ice-cold MeOH inside a −20 °C freezer. For PC2 and SEC8 staining, we used an IFM protocol method described in (He et al, 2018). All coverslips were mounted with 6% n-propyl gallate (Sigma-Aldrich, Cat #P3130) diluted in UltraPure Glycerol (Invitrogen, Cat #15514-001) and 10xPBS and combined with Epredia Immu-Mount (Epredia, Cat #9990402) in a 1:12 ratio.

Images of cells seeded on coverslips were obtained with an Olympus BX63 upright microscope equipped with a DP72 color, 12.8 megapixels, 4140 × 3096 resolution camera, and Olympus UPlanSApo 60× oil microscope objective. Images of the transwell filter-grown polarized epithelial cells were obtained with an Olympus IX83 inverted microscope, equipped with a Yokogawa CSU-W1 confocal scanner unit, ORCA-Flash4.0 V3 Digital CMOS camera (type number: C13440-20CU), and Olympus UPlanSApo 100x oil microscope objective. To prepare the images for publication, we used cellSens 1.18 software for constrained iterative deconvolution and assembled montages with Fiji and Adobe Photoshop 2023.

Live-cell imaging of RPE1 cells stably expressing SMO-tRFP and transiently expressing eGFP-DLG1 was done as described in (Juhl et al, 2023).

## Immunofluorescence staining of kidney sections

All animal studies were performed following the guidelines of the Institutional Animal Care and Use Committee at Washington University and the National Institutes of Health and approved by The Animal Studies Committee of Washington University (Approval No. 20100105). Animals are maintained in a barrier facility on the Washington University Medical Center campus. Husbandry services are provided by the facility whose dedicated staff monitor, feed and exchange cages on a fee-for-service basis. The mouse kidney specimens assayed for ciliary length, SDCCAG3 and IFT20 localization were obtained from $Pax3Cre-Dlg1^{F/F}$ mice and control (WT) littermates that were previously described (Kim et al, 2014b). For immunofluorescence staining of paraffin-embedded sections, antigen unmasking was performed by boiling the slides in antigen-retrieval buffer (10 mM Tris Base, 1 mM EDTA, and 0.05% Tween-20, pH 9.0) for 30 min. Samples were permeabilized with 0.05% Triton X-100 in PBS (PBS-T) for 10 min at room temperature, incubated in blocking buffer (3.0% BSA and 0.1% Triton X-100 in PBS) for 1 h, followed by staining with primary antibodies against SDCCAG3, IFT20 or acetylated tubulin overnight at 4 °C. After three washes with PBS-T, samples were incubated with secondary Alexa Fluor dye-conjugated antibodies for 1 h at room temperature. Nuclei were stained with DAPI, and specimens mounted using Mowiol containing n-propyl gallate (Sigma-Aldrich). Images were captured using a Nikon Eclipse Ti-E inverted confocal microscope equipped with a 60x Plan Fluor oil immersion (1.4 NA) and 100x Plan Fluor oil immersion (1.45 NA) objectives. A series of digital optical sections (z-stacks) were captured using a Hamamatsu ORCA-Fusion Digital CMOS camera at room temperature, and 3D image reconstructions were

produced. Images were processed and analyzed using Elements AR 5.21 (Nikon), Adobe Illustrator and Photoshop software. Cilia length measurements were calculated from control ($n = 437$ cilia, 3 mice) and *Pax3Cre-Dlg1$^{F/F}$* ($n = 252$ cilia from 4 mice). Ciliary levels of acetylated α-tubulin were derived from control ($n = 102$ cilia) and *Pax3Cre-Dlg1$^{F/F}$* ($n = 109$ cilia). Total ciliary levels of SDCCAG3 and IFT20 were measured per cilium, and the data were calculated from: SDCCAG3 ($n = 105$ cilia from 3 control mice and $n = 110$ from 4 *Pax3Cre-Dlg1$^{F/F}$* mice); IFT20 ($n = 99$ cilia from 3 control mice and $n = 104$ from 4 *Pax3Cre-Dlg1$^{F/F}$* mice).

## Quantitative real-time PCR (RT-qPCR)

Isolation of total RNA was performed using the NucleoSpin RNA II kit (Macherey-Nagel, Cat # 740955.50) following the manufacturer's instructions. RNA was reverse-transcribed using Superscript III Reverse Transcriptase (Invitrogen, Cat #18080-044) and cDNA amplified by qPCR using SYBR Green (Applied Biosystems, Cat #4309155). The qPCR was conducted in triplicate using the QuantStudio 7 Flex Real-Time PCR system with the following steps: 95 °C for 10 min, 40 cycles of [95 °C for 30 s, 60 °C for 1 min, 72 °C for 30 s], 95 °C for 1 min. Primer sequences used in this study for RT-qPCR are listed in the Reagents and Tools table. mRNA levels were determined using the comparative threshold cycle (Ct) method, normalizing to GAPDH and 18 S ribosomal RNA. The mRNA levels were expressed relative to that in WT mCCD cells.

## Inhibition of DLG1

Acute inhibition of DLG1 was done using the dimeric peptides AVLX-144 (YGRKKRRQRRR-*N*PEG$_4$(IETDV)$_2$, Tat-*N*-dimer), ReTat-*N*-dimer (rrrqrrkkr-*N*PEG$_4$(IETDV)$_2$ containing a retroinverso Tat cell-penetrating sequence and the non-PDZ-binding control AVLX-144-AA (YGRKKRRQRRR-*N*PEG$_4$(IEADA)$_2$ containing alanine mutations in the dimeric region (Bach et al, 2008; Bach et al, 2012). Compounds were purchased from WuXi AppTec (Shanghai, China) as hydrochloride salts and purities were checked by mass spectrometry. Prior to the inhibitor experiment, the WT mCCD cells were seeded on glass coverslips and allowed to reach 80% confluence. To promote ciliogenesis, the cells were subjected to a 24 h starvation period using the starvation medium outlined previously. After 12 h of starvation, the medium was changed to the inhibitor-supplemented starvation medium and incubated for an additional 12 h. Subsequently, the cells were examined using IFM analysis.

## Immunoprecipitation, SDS-PAGE, and western blot analysis

Immunoprecipitation in mCCD and HEK293T cells was carried out as described previously (Goncalves et al, 2021), except that the washing buffer contained 0.1% NP-40 instead of 0.5% NP-40. Input and pellet fractions were analyzed by SDS-PAGE and western blotting as described previously (Goncalves et al, 2021) by using antibodies and dilutions as listed in the Reagents and Tools table.

## TGFβ stimulation assay

Following cell seeding and 24 h incubation with starvation medium, the cells were stimulated with 2 ng/mL recombinant human TGF-β1 (R&D Systems, Cat # 240-B) diluted in the starvation medium for varying durations of 30, 60, 90, and 120 min or left untreated (0 min). The cells were later lysed for subsequent analysis using the aforementioned SDS-PAGE and western blotting. The antibodies and dilutions used for this analysis are listed in the Reagents and Tools table.

## Quantitative image and statistical analysis

Using IFM images and Fiji software (Schindelin et al, 2012), we measured cilium length, frequency, and relative mean fluorescence intensity (MFI) of relevant antibody-labeled antigens at the cilium or ciliary base in WT, *Dlg1$^{-/-}$* and rescue IMCD3 and mCCD lines. Non-blinding was done. Unless otherwise stated, the results were confirmed in at least three independent biological replicates. Statistical analyses were performed using GraphPad Prism 10.0.1. For manual quantification of ciliary staining intensities of fluorescent images, the background-corrected MFI was normalized to relevant control cells/ mice. The data was was cumulated and tested for Gaussian normality using either D'Agostino's K-squared test or Shapiro–Wilk test. If the data followed a normal distribution, the two-tailed, unpaired Student's *t*-test was used when comparing two groups, or one-way ANOVA followed by Tukey's multiple comparison tests was used for comparing more than two groups. If the data did not follow a normal distribution, the nonparametric Mann–Whitney test was used when comparing two groups, or the Kruskal–Wallis test with Dunn's multiple pairwise comparison tests was used for comparing more than two groups. All quantitative data are presented as mean ± standard deviation unless otherwise specified. Differences were considered significant when the *P*-value was <0.05. ns, not statistically significant; *$P < 0.05$; **$P < 0.01$; ***$P < 0.001$; ****$P < 0.0001$. Quantitative analysis of western blot data was done as described previously (Goncalves et al, 2021).

## Automated image analysis and primary cilia intensity measurements

PC2 (Fig. 5C,D) and SDCCAG3 (Fig. 5E,F) intensity levels were measured in spinning disk fluorescence microscopy 3D image stacks of transwells-cultured cells acquired from WT, *Dlg1$^{-/-}$*, and rescue mCCD cell lines in three independent experiments, with a total of 15–25 images and 431-739 cells/cilia analyzed per condition. To minimize any bias and ensure experimental reproducibility, all intensity measurements were performed by a fully automated MATLAB script reporting the mean fluorescence intensity of the protein of interest inside subregions of the identified primary cilia. The functional steps of the script are reported below. First, (1) Nucleus regions were automatically identified (DAPI channel, Gaussian filtering, background subtraction and global thresholding) and (2) primary cilia were accurately segmented (cilia marker channel, Gaussian filtering, and local thresholding) as the brightest 3D objects overlapping a nucleus region. Next, (3) primary cilium bases were identified as the closest cilium voxel to the center of mass of the corresponding nucleus region (assuming an outward growth of the cilia), and (4) primary cilium base regions were defined as the set of cilium voxels within a maximum (user-defined) geodesic distance to the corresponding base. Finally, (5) SDCCAG3 and PC2 channels mean intensities were individually measured and reported inside each primary cilium, primary cilium base region, and primary cilium body (whole cilium excluding the base region) after background

intensity correction (3D median filtered image subtraction). Example images are shown in Appendix Fig. S9. The script was developed for this project by the Danish Bioimaging Infrastructure Image Analysis Core Facility (DBI-INFRA IACFF) and is available on the GitHub repository at https://github.com/DBI-INFRA/PrimaryCiliaAnalyzer.

## Alpha Fold modeling of protein complexes

Structures of protein complexes shown in Fig. 7C and Appendix Fig. S8A, B were modeled using a local installation of Alphafold v2.1.0 (Evans et al, 2022; Jumper et al, 2021) using sequences for *Mus musculus* (Mm) or *Homo sapiens* (Hs) DLG1, SDCCAG3, IFT20, and IFT54. Predicted interacting areas were inspected for a low Predicted Alignment Error (PAE) score as the main indicator for confidence. All figures of protein structures were prepared using PyMOL v. 2.5 (Schrodinger LLC, https://pymol.org).

## Graphics

Figure 7D was created with BioRender.com.

## Data availability

The datasets and computer code produced in this study are available in the following databases: (1) Mass spectrometry data: the mass spectrometry proteomics data have been deposited to the ProteomeXchange Consortium via the PRIDE (Perez-Riverol et al, 2022) partner repository with the dataset identifier PXD051912; (2) fully automated MATLAB script: the code used for automated ciliary fluorescence analysis in this article is available on the GitHub repository at https://github.com/DBI-INFRA/PrimaryCiliaAnalyzer. The repository also contains a test image as well as detailed instructions in the script file; (3) original source data for the fluorescent images shown in Fig. 1E, Fig. 3A,E, Fig. 5A–C,E, and Fig. 6D,E: BioImage Archive, accession number S-BIAD1084.

The source data of this paper are collected in the following database record: biostudies:S-SCDT-10_1038-S44319-024-00170-1.

## Peer review information

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

## Acknowledgements

The authors thank Søren L Johansen and Maria S Holm for expert technical assistance, and Pernille MH Olesen, Julie KT Sørensen, Søren Bjerg, and Maaryah Iqbal for contributing to the initial stages of this project, and Rachel Giles for helpful discussions and for contributing results that ended up not being included. We are grateful to the Polycystic Kidney Disease Research Resource Consortium (U54DK126114) for providing the anti-PC2 (rabbit polyclonal) antibody, and to Eric Féraille, Greg Pazour, Christopher Westlake, Kay Oliver Schink, Carlo Rivolta, and Michael Taschner for other reagents. This work was supported by grants NNF18SA0032928 and NNF22OC0080406 from the Novo Nordisk Foundation (LBP, SFP), grant 2032-00115B and 3103-00177B from the Independent Research Fund Denmark (LBP, STC), grant R01-DK108005 from the National Institute of Diabetes and Digestive and Kidney Diseases (MRM), Innovation grant 20OI174 from the Dutch Kidney Foundation (RR), the European Union's Horizon 2020 research and innovation program Marie Sklodowska-Curie Innovative Training Networks (ITN) grant 861329 (RR, RBR, KB, STC, LBP), and the Wellcome Trust grant 201585/B/18/Z (GDD, RBR).

## Author contributions

**Csenge K Rezi**: Formal analysis; Validation; Investigation; Visualization; Methodology; Writing—original draft; Writing—review and editing. **Mariam G Aslanyan**: Formal analysis; Supervision; Investigation; Methodology; Writing—review and editing. **Gaurav D Diwan**: Formal analysis; Visualization; Writing—original draft; Writing—review and editing. **Tao Cheng**: Formal analysis; Investigation; Visualization. **Mohamed Chamlali**: Formal analysis; Validation; Investigation; Visualization. **Katrin Junger**: Investigation; Methodology. **Zeinab Anvarian**: Supervision; Investigation; Methodology. **Esben Lorentzen**: Formal analysis; Visualization; Writing—original draft; Writing—review and editing. **Kleo B Pauly**: Formal analysis; Investigation. **Yasmin Afshar-Bahadori**: Formal analysis; Investigation. **Eduardo FA Fernandes**: Resources; Methodology; Writing—original draft; Writing—review and editing. **Feng Qian**: Resources. **Sébastien Tosi**: Software; Formal analysis; Visualization; Methodology. **Søren T Christensen**: Funding acquisition; Visualization. **Stine F Pedersen**: Funding acquisition; Writing—original draft. **Kristian Strømgaard**: Resources; Methodology. **Robert B Russell**: Supervision; Funding acquisition. **Jeffrey H Miner**: Resources; Supervision. **Moe R Mahjoub**: Formal analysis; Supervision; Funding acquisition; Visualization; Writing—original draft; Writing—review and editing. **Karsten Boldt**: Formal analysis; Supervision; Funding acquisition. **Ronald Roepman**: Supervision; Funding acquisition; Writing—review and editing. **Lotte B Pedersen**: Conceptualization; Supervision; Funding acquisition; Writing—original draft; Project administration; Writing—review and editing.

Source data underlying figure panels in this paper may have individual authorship assigned. Where available, figure panel/source data authorship is listed in the following database record: biostudies:S-SCDT-10_1038-S44319-024-00170-1.

## Disclosure and competing interests statement

The authors declare no competing interests.

