## [Peer Review File · EMBO Reports]

DLG1 functions upstream of SDCCAG3 and IFT20 to control ciliary targeting of polycystin-2

Csenge Rezi, Mariam Aslanyan, Gaurav Diwan, tao cheng, Mohamed Chamlali, Katrin Junger, Zeinab Anvarian, Esben Lorentzen, Kleo Pauly, Yasmin Afshar-Bahadori, Eduardo Fernandes, Feng Qian, Sébastien Tosi, Soren Christensen, Stine Pedersen, Kristian Stromgaard, Rob Russell, Jeffrey Miner, Moe Mahjoub, Karsten Boldt, Ronald Roepman, and Lotte Pedersen

Corresponding author(s): Lotte Pedersen (lpedersen@bio.ku.dk)

Review Timeline:

Submission Date:	11th Dec 23
Editorial Decision:	25th Jan 24
Revision Received:	14th Mar 24
Editorial Decision:	19th Apr 24
Revision Received:	8th May 24
Accepted:	29th May 24

Editor: Martina Rembold / Deniz Senyilmaz Tiebe

Transaction Report:

Dear Prof. Pedersen

Thank you for the submission of your research manuscript to our journal and thank you for your feedback on the referee reports we received. These are copied again below.

The referees consider the findings potentially interesting but also raise a number of concerns that need to be addressed in a revised version. Referee 1 and 3 also raised concerns regarding the link between DLG1 and kidney cyst formation and CAKUT. Upon further feedback, referee 2 agreed with these comments. In particular, referee 1 is concerned that the hydronephrosis phenotype you observe with the Pax3:Cre line might secondarily affect cell polarization and ciliary length and therefore considered it important to confirm the *in vivo* data with an independent Cre line. The referee proposed to obtain material from Jeffrey Miner's lab, which was a good proposal but you informed me that there is no material left from the DLG1 work and therefore mice would have to be derived from frozen sperm. Given that such mouse work would certainly take more than a year, I agree with your suggestion to put less focus on the mouse results, to carefully phrase your conclusions on kidney cysts and cilia and to transparently state and discuss the limitations of the current data set and mouse model used. Taken together, we invite you to revise your study along the lines you proposed in your feedback, addressing all referee concerns apart from the additional mouse work with an independent Cre line.

Please address all referee concerns in a complete point-by-point response. Acceptance of the manuscript will depend on a positive outcome of a second round of review. It is EMBO Reports policy to allow a single round of revision only and acceptance or rejection of the manuscript will therefore depend on the completeness of your responses included in the next, final version of the manuscript.

We realize that it is difficult to revise to a specific deadline. In the interest of protecting the conceptual advance provided by the work, we recommend a revision within 3 months (April 25th). Please discuss the revision progress ahead of this time with the editor if you require more time to complete the revisions.

I am also happy to discuss the revision further via e-mail or a video call, if you wish.

*****IMPORTANT NOTE:

We perform an initial quality control of all revised manuscripts before re-review. Your manuscript will FAIL this control and the handling will be delayed IN CASE the following APPLIES:

- 1) A data availability section providing access to data deposited in public databases is missing. If you have not deposited any data, please add a sentence to the data availability section that explains that.
- 2) Your manuscript contains statistics and error bars based on $n=2$. Please use scatter blots in these cases. No statistics should be calculated if $n=2$.

When submitting your revised manuscript, please carefully review the instructions that follow below. Failure to include requested items will delay the evaluation of your revision.*****

- 1) a .docx formatted version of the manuscript text (including legends for main figures, EV figures and tables). Please make sure that the changes are highlighted to be clearly visible.
- 2) individual production quality figure files as .eps, .tif, .jpg (one file per figure). Please download our Figure Preparation Guidelines (figure preparation pdf) from our Author Guidelines pages <https://www.embopress.org/page/journal/14693178/authorguide> for more info on how to prepare your figures.
- 3) a .docx formatted letter INCLUDING the reviewers' reports and your detailed point-by-point responses to their comments. As part of the EMBO Press transparent editorial process, the point-by-point response is part of the Review Process File (RPF), which will be published alongside your paper.
- 4) a complete author checklist, which you can download from our author guidelines (<<https://www.embopress.org/page/journal/14693178/authorguide>>). Please insert information in the checklist that is also reflected in the manuscript. The completed author checklist will also be part of the RPF.
- 5) Please note that all corresponding authors are required to supply an ORCID ID for their name upon submission of a revised manuscript (<<https://orcid.org/>>). Please find instructions on how to link your ORCID ID to your account in our manuscript

tracking system in our Author guidelines

(<<https://www.embopress.org/page/journal/14693178/authorguide#authorshipguidelines>>)

6) We replaced Supplementary Information with Expanded View (EV) Figures and Tables that are collapsible/expandable online. A maximum of 5 EV Figures can be typeset. EV Figures should be cited as 'Figure EV1, Figure EV2' etc... in the text and their respective legends should be included in the main text after the legends of regular figures.

7) Before submitting your revision, primary datasets (and computer code, where appropriate) produced in this study need to be deposited in an appropriate public database (see <<https://www.embopress.org/page/journal/14693178/authorguide#dataavailability>>). Specifically, we would kindly ask you to provide public access to mass spec experiment.

The accession numbers and database should be listed in a formal "Data Availability " section (placed after Materials & Method) that follows the model below (see also <<https://www.embopress.org/page/journal/14693178/authorguide#dataavailability>>). Please note that the Data Availability Section is restricted to new primary data that are part of this study.

Data availability

Additional information on source data and instruction on how to label the files are available <<https://www.embopress.org/page/journal/14693178/authorguide#sourcedata>>.

10) Figure legends and data quantification:

- the name of the statistical test used to generate error bars and P values,
- the number (n) of independent experiments (please specify technical or biological replicates) underlying each data point,
- the nature of the bars and error bars (s.d., s.e.m.)
- If the data are obtained from n {less than or equal to} 5, show the individual data points in addition to the SD or SEM.
- If the data are obtained from n {less than or equal to} 2, use scatter blots showing the individual data points.

11) Our journal encourages inclusion of *data citations in the reference list* to directly cite datasets that were re-used and obtained from public databases. Data citations in the article text are distinct from normal bibliographical citations and should directly link to the database records from which the data can be accessed. In the main text, data citations are formatted as follows: "Data ref: Smith et al, 2001" or "Data ref: NCBI Sequence Read Archive PRJNA342805, 2017". In the Reference list, data citations must be labeled with "[DATASET]". A data reference must provide the database name, accession number/identifiers and a resolvable link to the landing page from which the data can be accessed at the end of the reference. Further instructions are available at <<https://www.embopress.org/page/journal/14693178/authorguide#referencesformat>>.

12) All Materials and Methods need to be described in the main text. We would encourage you to use 'Structured Methods', our new Materials and Methods format. According to this format, the Materials and Methods section should include a Reagents and Tools Table (listing key reagents, experimental models, software and relevant equipment and including their sources and relevant identifiers) followed by a Methods and Protocols section in which we encourage the authors to describe their methods using a step-by-step protocol format with bullet points, to facilitate the adoption of the methodologies across labs. More information on how to adhere to this format as well as downloadable templates (.doc or .xls) for the Reagents and Tools Table can be found in our author guidelines: <<https://www.embopress.org/page/journal/14693178/authorguide#manuscriptpreparation>>. An example of a Method paper with Structured Methods can be found here: <<https://www.embopress.org/doi/10.15252/msb.20178071>>.

13) As part of the EMBO publication's Transparent Editorial Process, EMBO Reports publishes online a Review Process File to accompany accepted manuscripts. This File will be published in conjunction with your paper and will include the referee reports, your point-by-point response and all pertinent correspondence relating to the manuscript.

Kind regards,

Referee #1:

The at hand manuscript by Rezi et al., published already as a preprint by Nov 2023, aims to analyze the function of DLG1, a main component of the Scribble polarity complex, in ciliated renal epithelial cells. The authors describe a role of DLG1 in modulating ciliary protein trafficking and the cilia proteome composition. They demonstrate, that Pax3:cre based knockout of Dlg1 results in hydronephrosis and elongation of primary cilia, while in cells the knockout of DLG1 altered ciliary protein composition as measured by proximity labeling and length when cultured on transwell dishes. These data suggest that DLG1 is required for the ciliary targeting of SDCCAG3 and IFT20, proteins that were previously implicated with the ciliary targeting of PC2. Subsequent experiments show that ciliary targeting of PC2 also depends on DLG1. There is also an important link between DLG1 and CAKUT (Congenital Anomalies of the Kidney and Urinary Tract) and wildtype but not DLG1 with a CAKUT-associated variant could rescue the cilia phenotype in Dlg^{-/-} cells.

The at-hand study is on an important topic within cell biology and nephrology. It uncovers a novel mechanism regulating ciliary expression of PC2 and ciliary function and confirms the importance of Dlg1 for kidney homeostasis and the maintenance of the tissue architecture. I really like the data and the story, but there are some concerns that should be addressed by the authors:

1. Most importantly, I do not see cyst formation or precystic tubular dilations, as stated by the authors, in the pictures presented in Fig 1A/B. There is the very prominent phenotype of hydronephrosis which is most likely due to defects in the ureter - also explained by Pax3 expression down that road. Therefore, to point out the role of DLG1 for cyst formation and to link Dlg1 to

renal ciliopathies and cystic kidney disease, the authors should use another cre line for confirmation: one that exclusively expresses cre in the proximal or distal nephron, without leading to hydronephrosis by affecting the ureter. This material might be available since such crossings had been done in Jeff Miner's lab that generated the transgenic mice (PMID: 24699546). It will be essential to investigate ciliary length in kidneys of those mice and confirm this key finding in vivo.

2. In addition, the occurrence of hydronephrosis might alter the entire flow and mechanics within the kidney significantly, which could secondarily affect cell polarization and ciliary length. Therefore it would be important to do the ciliary length analysis at a time point when the kidney is still unaffected (or in a different cre background). How many animals were investigated for Fig 1B and of which exact age? The authors should further distinguish between proximal and distal tubules by using different markers and analyze tubular diameter in these kidneys.

Minor points:

1. AD-PKD should be ADPKD (line 52 etc)
2. The conclusion of the introduction nearly replicates the abstract's final text verbatim. Although it's a minor issue, I prefer not to read the same words and content twice.
3. The authors could be clearer on the fact that the renal phenotype of these mice has already been described in 2014.
4. Re Fig S1: What is the localization of endogenous DLG1 in RPE cells? Is it ciliary as well?
5. The term "physical interaction" when describing co-precipitations from HEK cells is not ideal since this is suggestive of direct interactions. In the COIPs, the proteins are part of common complexes, but do not necessarily directly interact. I would suggest rephrasing this or performing in vitro interaction experiments using recombinant purified proteins.

Referee #2:

In this manuscript, the authors use state of the art technologies to analyze the function of DLG1. The authors show that deletion of the Dlg1 gene in the kidneys causes elongation of cilia. A similar phenotype was observed in fully polarized mCCD Dlg1 knock out cells, but not in non-polarized cells. Interestingly, also the localization of DLG1 differed. It was found at the lateral membrane in fully polarized cells, but at the base of and in cilia in RPE cells.

Using cilium specific proximity labeling in IMCD3 cells the authors next identified proteins either lost from cilia, or enriched in cilia in Dlg1 knockout cells. They focus subsequently on SDCCAG3 and IFT20 and confirm Dlg1 knockout results in less of these proteins in cilia or at the base of cilia indicating that DLG1 plays a role in trafficking of SDCCAG3 and IFT20 (in)to cilia. These analyses lead the authors to test whether PC2 levels are affected in Dlg1 knockout cells, and indeed they find less PC2 in the cilia of knockout cells or in cells in which DLG1 is inhibited. To test if this could also play a role in CAKUT patients, they subsequently tried to rescue the defects using either a wt or a Dlg1 mutant variant, which shows that the disease variant cannot rescue whereas the wt gene can. Finally, the authors show that DLG1 interacts with SDCCAG3 and IFT20, using overexpressed tagged proteins in IP experiments and alpha fold based modeling.

The paper is well written, and results have been clearly presented.

Major comments

The manuscript described results with various cells: mouse kidney, mCCD, RPE and IMCD3 cells. This is a bit confusing as the effects are different between the different cells. Also the localization of DLG1 is different in different cells, but I think its localization in the IMCD3 cells has not been analyzed here. I think this is important information as these cells were used to identify cilia proteome changes upon DLG1 depletion. Please analyze DLG1 localization in IMCD3 cells.

The authors test whether the DLG1T507R variant can rescue the localization of SDCCAG3 and IFT20 in Dlg1 knockout cells. Ift20 levels cannot be rescued. The authors show a small but significant effect on SDCCAG3 levels, suggesting there is some rescue. In the text they state there is no rescue, while the wt does rescue, shown in Fig 3. To substantiate the claim that the DLG1T507R variant shows less rescue than the wt DLG, the authors should repeat these experiments, testing the two constructs side by side and use statistical analysis to test whether there is a significant difference. The results available now do not allow such a conclusion.

Figure 4A shows anti-acetylated tubulin staining of wt and Dlg1 knockout mouse kidney sections. I was surprised by the big difference in acetylated tubulin levels between the wt and knockout. I could not find a remark about this in the manuscript. Please discuss this.

In the discussion, on line 333-334, the authors state: "Reduced ciliary presence of polycystins may at least be partly responsible for the observed ciliary length phenotype of DLG1-deficient cells". I don't see how PC levels would cause changes in ciliary length. Has this been shown?

Minor remarks

Line 53, please change "is" into "can be", as also mutations in other gene scan cause ADPKD.

As the authors use various cell lines in their analyses, it would be very helpful if they could also indicate in the figures

themselves which cells were used. E.g. Fig 3A-D report on IMCD3 cells, whereas Fig3E-G are about mCCD cells. Please do so for all figures.

Fig 6J lacks statistical analysis. Please include this.

I found the conclusion sentence about the interactions between DLG1, SDCCAG3 and IFT20 a bit too strong. These are in vitro experiments with overexpressed proteins and modeling. Please tune down this conclusion a bit and indicate that it is based on in vitro experiments and modeling.

The results section ends with two very similar conclusion sentences. Please correct this.

There were some English errors:

Line 147. Please add "of" between "amount" and "Dlg3".

Line 297, Please remove "in" at the end of the sentence.

Line 420, please change "we" into "were".

Referee #3:

In this manuscript, Rezi et al. identify Dlg1 as a novel regulator of primary cilia. Dlg1 has known roles as a Scribble complex component in establishing basolateral polarity but prior studies have also suggested a link between cilia and Dlg1 on the basis of its localization to the centrosome in lung cancer cells and presence in the ciliary proteome of inner medullary collecting duct (IMCD3) cells. In their study, Rezi et al. assess the functional role of Dlg1 in kidney cilia, finding that conditional Dlg1 knockout in mouse kidney leads to structural defects at the organ level, as well as increased ciliary length and altered ciliary protein composition. Using cilia proximity labeling-based proteomic analysis in IMCD3 Dlg1 KO cells, the authors identified ciliary proteins that may be regulated by Dlg1, many of which are associated with proper cilia formation and maintenance as well as ciliary signaling. The authors focused on two hits from their proteomic screen: the retromer-associated protein SDCCAG3 and the IFT-B complex component IFT20, both of which have been previously implicated in trafficking of PKD2 to the cilium. Indeed, Dlg1 KO led to diminished SDCCAG3 localization to cilia in vivo and in IMCD3 cells, and to the ciliary base in mCCD cells. IFT20 protein levels were similarly reduced at the ciliary base in vivo and in mCCD cells. Furthermore, PC2 localization to the cilia was reduced in Dlg1 KO mCCD cells. The authors further note that a reported patient-derived Dlg1 mutant associated with congenital anomalies of the kidney and urinary tract (CAKUT) disorder failed to rescue IFT20 and SDCCAG3 localization in Dlg1 KO cells, hinting at a possible clinical connection between CAKUT and cilia. Lastly, colPs and structural modeling suggested SDCCAG3 and IFT20 as interacting partners with each other and Dlg1 (albeit potentially indirect), strengthening the authors' Dlg1-dependent SDCCAG3 and IFT20 trafficking model. Taken together, these findings identify Dlg1 as a new potential ciliary regulator and offer some additional mechanistic insight (via IFT20 and SDCCAG3) and physiological relevance into Dlg1's ciliary role. Noted below are key questions that need to be addressed regarding some of the reported phenotypes and the potential connection to CAKUT. If the authors are able to clarify these issues, I believe their study would be of interest in the field and appropriate for EMBO Reports's audience.

Major points

1. What is the evidence for DLG1-T489R having any association with CAKUT? This mutant is not obviously listed in ref. 34 that is cited in the manuscript. This is an important point given that the use of this allele and the claimed relevance to CAKUT rest on it. Additionally, in the canonical DLG1 sequence (Uniprot Q12959), residue 489 is not a threonine. Clarification of the DLG1 sequence and which associated isoforms are expressed in human/mouse kidneys would be valuable.
2. It would be helpful for the authors to discuss any prior evidence that might suggest a link between cilia and CAKUT. The authors suggest that Dlg1-linked ciliary alterations may be responsible for CAKUT, but is it not also possible that Dlg1 separately impacts cilia and CAKUT?
3. In the abstract, the authors report 'cystogenesis' in Dlg1 mutant kidney, but the data to support this seem scant. The Results report 'tubular dilations that appeared to be pre-cystic' but these are not rigorously or quantitatively characterized.
4. While the authors claim that Dlg1 regulates ciliary length indirectly as a principally plasma membrane-localized protein, they also note that over-expressed GFP-Dlg1 localized to the cilium base in RPE1 cells. Did the authors analyze the localization of overexpressed Dlg1-mCh in mCCD and IMCD3 cells (and not just RPE1 cells)? Conversely, what is the localization of endogenous Dlg1 in RPE1 cells?
5. The authors claim that SDCCAG3 is a retromer subunit, but is this true? Evidence seems to indicate it can be identified via IP/MS as a Vps35-associated protein (ref. 41), but not necessarily as a bona fide subunit of retromer. In the absence of further evidence that SDCCAG3 is part of a bona fide retromer complex, it would seem more appropriate to call it a retromer-associated protein or binding partner.
6. It is a bit odd that there is a cell-type-specific effect of DLG1 KO on IFT20 localization - IFT20 is reduced in mCCD Dlg1 KO cells but not in mutant IMCD3 cells, even though mutant IMCD3 cells were used in the proximity labeling experiments that first identified an effect of Dlg1 KO on ciliary IFT20 levels. It would be helpful if the authors provided more of an explanation for this apparent discrepancy.
7. In Fig. 5 (and possibly 3E), the contrast of the microscopy images appears to be saturated, precluding a direct comparison of

the intensity of SDCCAG3 and PC2 levels across the different experimental conditions. Despite this, in Fig 5C and 5E, the PC2 and SDCCAG3 intensity and ciliary localization do not look to be restored to wildtype-like status upon expression of mCh-DLG1, even the the quantifications do indicate full rescue. Can the authors clarify this issue? How exactly were signals quantified?

8. In Fig. 6F, the basal body levels of IFT20 appear to be significantly lower in cells expressing the T507R mutant than in DLG1 KO cells. Can the authors comment on this or suggest an explanation?

9. What is the effect of the DLG1-T507R on ciliary PC2 levels?

10. Alpha Fold reported identification of a "high confidence interaction between IFT20 and SDCCAG3" (lines 308-309). What is the evidence that the interaction is "high confidence"? The wording in the abstract "using biochemical approaches and AlphaFold modelling we show that DLG1 associates physically with SDCCAG3 and IFT20" is perhaps confusing/misleading as it arguably implies that Alpha Fold predicted interactions between Dlg1 and IFT20, or between Dlg1 and SDCCAG3, which apparently was not the case.

Minor points

1. Why is it important that Pax3Cre is active in urogenital mesenchyme (line 135)? The relevance of this to the present findings is unclear.
2. In line 172 of the manuscript, the text refers to using Arl13b as the ciliary marker in Fig. S2B, but AcTub is actually shown. Either the figure or the text should be corrected.
3. Not only are cilia longer in Dlg1 knockouts versus WT, but the intensity of the ciliary marker AcTub appears to be greatly increased in vivo (see Fig. 1B, 4A). Is this in fact the case or observed in cell lines?
4. Could the authors provide a comment on the shape of the volcano plot and why multiple hits having apparently maximal magnitudes of enrichment/depletion (albeit different p-values)? Additionally, why is Dlg1 not more strongly depleted?
5. In line 210 of the manuscript, reference 15 appears to show SDCCAG3 localization to the basal body/centrioles rather than to the primary cilia.
6. In Fig. 3B and 3F, significance tests should be added comparing SDCCAG3 localization between WT v. Dlg1^{-/-} mCh-Dlg1.
7. What is quantified in Fig. 4B? Is the fluorescence intensity the average across many cells/cilia or from different samples/kidneys analyzed?
8. Asterisks indicating the ciliary base are missing from Fig. 5C and 5E.
9. For Fig. S5, what would happen if the ALVX-144 and ReTAT-N-dimer peptides are combined? Would there be a more drastic drop in PC2 levels at the cilia/centrosome? Additionally, it would be informative to include significance values comparing 0uM and 25uM ReTat-N-dimer in Fig. S5B (top panel). Could the authors comment on why treating cells with a 25uM concentration of ReTat-N-dimer results in a less significant result than 5uM in Fig. S5B (bottom panel)?
10. For Fig. 5D, the P values between comparing PC2 levels in WT v. Dlg1^{-/-} mCh-Dlg1 seem to indicate highly significant differences. Are these correct, which would suggest incomplete rescue by the transgene?
11. Fig 5C. The green channel appears to be missing from top left sub-panel
12. Are the regions of IFT20/SDCCAG3 predicted by Alpha Fold to mediate their association also functionally required for a co-IP between these two proteins?

We thank all the referees for their insightful and constructive comments and have done our best to address them as described in our point-by-point responses below.

Referee #1

The at hand manuscript by Rezi et al., published already as a preprint by Nov 2023, aims to analyze the function of DLG1, a main component of the Scribble polarity complex, in ciliated renal epithelial cells. The authors describe a role of DLG1 in modulating ciliary protein trafficking and the cilia proteome composition. They demonstrate, that Pax3:cre based knockout of Dlg1 results in hydronephrosis and elongation of primary cilia, while in cells the knockout of DLG1 altered ciliary protein composition as measured by proximity labeling and length when cultured on transwell dishes. These data suggest that DLG1 is required for the ciliary targeting of SDCCAG3 and IFT20, proteins that were previously implicated with the ciliary targeting of PC2. Subsequent experiments show that ciliary targeting of PC2 also depends on DLG1. There is also an important link between DLG1 and CAKUT (Congenital Anomalies of the Kidney and Urinary Tract) and wildtype but not DLG1 with a CAKUT-associated variant could rescue the cilia phenotype in Dlg^{-/-} cells.

The at-hand study is on an important topic within cell biology and nephrology. It uncovers a novel mechanism regulating ciliary expression of PC2 and ciliary function and confirms the importance of Dlg1 for kidney homeostasis and the maintenance of the tissue architecture. I really like the data and the story, but there are some concerns that should be addressed by the authors:

****We thank the referee for an insightful and generally positive evaluation of our manuscript.**

1. Most importantly, I do not see cyst formation or precystic tubular dilations, as stated by the authors, in the pictures presented in Fig 1A/B. There is the very prominent phenotype of hydronephrosis which is most likely due to defects in the ureter - also explained by Pax3 expression down that road. Therefore, to point out the role of DLG1 for cyst formation and to link Dlg1 to renal ciliopathies and cystic kidney disease, the authors should use another cre line for confirmation: one that exclusively expresses cre in the proximal or distal nephron, without leading to hydronephrosis by affecting the ureter. This material might be available since such crossings had been done in Jeff Miner's lab that generated the transgenic mice (PMID: 24699546). It will be essential to investigate ciliary length in kidneys of those mice and confirm this key finding in vivo.

****We appreciate the referee's insightful comments. Unfortunately, it will not be possible for us to do additional experiments using another Cre line as suggested. The Pax3Cre-Dlg1^{F/F} KO kidneys that we analyzed for our manuscript are the only tissues that remain from the prior work on Dlg1 in Jeffrey Miner's lab. There are no mice available. We would have to re-derive mice using frozen sperm, then setup crosses to get the KO genotype, then isolate kidneys for analysis etc., which could easily take more than a year to do, just to derive the mice. Unfortunately we do not currently have the resources to initiate this work, and we thank the referee in advance for understanding. We have done our best to address (in writing and with experiments) the issues about the Pax3Cre-Dlg1^{F/F} results pointed out by the referee.**

Importantly, we would like to emphasize that the Pax3Cre-Dlg1^{F/F} mice do develop tubular dilations and cysts in the nephron of the cortical regions of the kidney,

independent of hydronephrosis. In addition to the urogenital mesenchyme, *Pax3Cre* is expressed in the metanephric mesenchyme, which differentiates to form the various segments of the mature nephron (glomerulus, proximal tubule, loop of Henle, and distal tubule). Therefore, *Pax3Cre-Dlg1^{F/F}* mice result in loss of Dlg1 expression in all nephron (but not ureteric bud) epithelial cell derivatives. We have now included magnified images of the kidney cortex from *Pax3Cre-Dlg1^{F/F}* mice to highlight these cystic dilations (Figure 1A). Moreover, this cystic phenomenon was previously observed by the Miner lab upon ablation of Dlg1 in the metanephric mesenchyme using Six2-Cre (Ahn et al, 2013), which is also expressed in the metanephric mesenchyme, and thus leads to deletion of Dlg1 in the same nephron segments as Pax3Cre. We have included the relevant figure from the Ahn et al (2013) study below for the reviewer. In the past, we also examined cilia length in the Six2-Cre-Dlg1 KO mice from the Ahn 2013 study. Although this data was not published, we have included them here to demonstrate to the reviewer that cilia were similarly elongated in those cystic tubules, independent of hydronephrosis (note- those animals do not develop hydronephrosis). Finally, we show that loss of Dlg1 in vitro also causes changes in cilia length (see Figure 1E-G). Therefore, three independent lines of evidence, both in vitro and in vivo, demonstrate that the observed changes in cilia length are due to loss of Dlg1 function and not simply a secondary effect of hydronephrosis. We have added these details to the Results section.

2. In addition, the occurrence of hydronephrosis might alter the entire flow and mechanics within the kidney significantly, which could secondarily affect cell polarization and ciliary length. Therefore it would be important to do the ciliary length analysis at a time point when the kidney is still unaffected (or in a different cre background).

****As described in point 1, we do not believe the changes in cilia length are due to secondary effects of hydronephrosis. There are in vivo and in vitro data that support our conclusion.**

How many animals were investigated for Fig 1B and of which exact age? The authors should further distinguish between proximal and distal tubules by using different markers and analyze tubular diameter in these kidneys.

****We analyzed N=4 *Pax3Cre-Dlg1^{F/F}* mice and N=3 littermate controls. Their ages were 4 months. We have added these details to the Figure 1 legend. Although we can try and co-stain with segment markers to define the origin of the cysts, we do not believe this would add any important information, as the expression of Pax3Cre is well-defined and thus both segments of the nephron are likely impacted.**

Minor points:

1. AD-PKD should be ADPKD (line 52 etc)

****This has now been corrected.**

2. The conclusion of the introduction nearly replicates the abstract's final text verbatim. Although it's a minor issue, I prefer not to read the samewords and content twice.

****We apologize for this and have now updated the Summary and Introduction to avoid replication of text.**

3. The authors could be clearer on the fact that the renal phenotype of these mice has already been described in 2014.

****We have added details of the phenotypes of the mice from previous studies in the Introduction and Results sections.**

4. Re Fig S1: What is the localization of endogenous DLG1 in RPE cells? Is it ciliary as well?

****We have unpublished data from cilia proteomics analyses indicating that DLG1 is indeed present in cilia of RPE1 cells, in line with the eGFP-DLG1 overexpression results shown in Figure S1E. We have not rigorously analyzed cilia localization of endogenous DLG1 by IFM in RPE1 cells given that our main focus was on kidney epithelial cells, and since extensive IFM analysis failed to show reproducible cilia localization of endogenous DLG1 in mCCD or IMCD3 cells (however see the response to referee #2's first comment below). We believe our inability to detect endogenous or tagged DLG1 at cilia in mCCD and IMCD3 cells may be due to technical reasons and/or because DLG1 only localizes to cilia under certain conditions. We now mention this in the Results section of the revised manuscript.**

5. The term "physical interaction" when describing co-precipitations from HEK cells is not ideal since this is suggestive of direct interactions. In the CO-IPs, the proteins are part of common complexes, but do not necessarily directly interact. I would suggest rephrasing this or performing in vitro interaction experiments using recombinant purified proteins.

****We have rephrased the relevant sentences as suggested.**

Referee #2

In this manuscript, the authors use state of the art technologies to analyze the function of DLG1. The authors show that deletion of the Dlg1 gene in the kidneys causes elongation of cilia. A similar phenotype was observed in fully polarized mCCD Dlg1 knock out cells, but not in non-polarized cells. Interestingly, also the localization of DLG1 differed. It was found at the lateral membrane in fully polarized cells, but at the base of and in cilia in RPE cells.

Using cilium specific proximity labeling in IMCD3 cells the authors next identified proteins either lost from cilia, or enriched in cilia in Dlg1 knockout cells. They focus subsequently on SDCCAG3 and IFT20 and confirm Dlg1 knockout results in less of these proteins in cilia or at the base of cilia indicating that DLG1 plays a role in trafficking of SDCCAG3 and IFT20 (in)to cilia. These analyses lead the authors to test whether PC2 levels are affected in Dlg1 knockout cells, and indeed they find less PC2 in the cilia of knockout cells or in cells in which DLG1 is inhibited. To test if this could also play a role in CAKUT patients, they subsequently tried to rescue the defects using either a wt or a Dlg1 mutant variant, which shows that the disease variant cannot rescue whereas the wt gene can. Finally, the authors show that DLG1 interacts with SDCCAG3 and IFT20, using overexpressed tagged proteins in IP experiments and alpha fold based modeling.

The paper is well written, and results have been clearly presented.

****We thank the referee for a constructive and generally positive evaluation of our manuscript.**

Major comments

The manuscript described results with various cells: mouse kidney, mCCD, RPE and IMCD3 cells. This is a bit confusing as the effects are different between the different cells. Also the localization of DLG1 is different in different cells, but I think its localization in the IMCD3 cells has not been analyzed here. I think this is important information as these cells were used to identify cilia proteome changes upon DLG1 depletion. Please analyze DLG1 localization in IMCD3 cells.

****We apologize for the cell line confusion. We only included one image of RPE1 cells transiently overexpressing eGFP-DLG1 (Figure S1E) since this was the only condition where we observed reproducible cilia localization of DLG1. We believe that our use of two different kidney epithelial cell line (mCCD and IMCD3) is a strength rather than weakness of our manuscript, since it allowed us to demonstrate that the effect of DLG1 loss on cilia composition is not cell type-specific but can be observed in at least two kidney epithelial cell types and *in vivo*.**

We have performed extensive IFM analysis for endogenous as well as tagged versions of DLG1 in both mCCD and IMCD3 cells (using various fixations and antibodies, stable and overexpressed exogenous DLG1 etc.), but have failed to reproducibly detect DLG1 at cilia in these cells. In one experiment, we were able to detect endogenous DLG1 at cilia of IMCD3 cells, as shown in the figure below. However, since we were unable to repeat this result, despite multiple attempts, we do not feel comfortable including these images in the manuscript. We believe our inability to detect endogenous or tagged DLG1 at cilia in mCCD and IMCD3 cells may be due to technical reasons and/or because DLG1

only localizes to cilia under certain conditions. We now mention this in the Results section of the revised manuscript.

Figure: IFM analysis of endogenous DLG1 localization in WT and *Ift27*^{-/-} IMCD3 cells. Proteins were stained with the indicated antibodies and visualized with IFM. DLG1 mainly localized to the cell edges and the base of cilia in IMCD3 WT, though the image shown in A also shows DLG1 along the cilia length. No major effect of *Ift27*^{-/-} on DLG1 localization was observed (B). The square highlights the location of the cilia and the arrows indicate DLG1 localization. Based on one experiment.

The authors test whether the DLG1T507R variant can rescue the localization of SDCCAG3 and IFT20 in Dlg1 knockout cells. Ift20 levels cannot be rescued. The authors show a small but significant effect on SDCCAG3 levels, suggesting there is some rescue. In the text they state there is no rescue, while the wt does rescue, shown in Fig 3. To substantiate the claim that the DLG1T507R variant shows less rescue than the wt DLG, the authors should repeat these experiments, testing the two constructs side by side and use statistical analysis to test whether there is a significant difference. The results available now do not allow such a conclusion.

****Thank you for this comment. We have now included data for the WT DLG1 rescue construct in the comparative quantitative IFM analysis of IFT20 and SDCCAG3 ciliary levels of our mCCD cell lines (new Figure 6F and G). Please note that we had actually included this cell line in our original IFM analysis but did not show the data in the old version of the**

manuscript. Therefore, the quantitative data for WT, *Dlg1*^{-/-}, and the mCh-DLG1^{T507R} rescue line is the same as before. The new quantitative analysis clearly shows that ciliary levels of SDCCAG3 and IFT20 are fully rescued by the WT DLG1 construct, but not by the DLG1^{T507R} variant. In addition, we now also measured ciliary length in all four mCCD cell lines grown on transwell filters, and found that the DLG1^{T507R} variant is unable to restore ciliary length in the *Dlg1*^{-/-} line to normal (new Figure 6C).

Figure 4A shows anti-acetylated tubulin staining of wt and *Dlg1* knockout mouse kidney sections. I was surprised by the big difference in acetylated tubulin levels between the wt and knockout. I could not find a remark about this in the manuscript. Please discuss this.

****The increase in the acetylated tubulin staining is due to the fact that the cilia are longer, and thus there is "more" overall signal for acetylated tubulin. We have now quantified the levels of acetylated tubulin in control and *Dlg1* KO kidneys (new Figure 1D). As expected, there is an increase in total acetylated tubulin intensity per cilium, consistent with the increase in ciliary length (Figure 1C). This information has been added to the Results section.**

In the discussion, on line 333-334, the authors state: "Reduced ciliary presence of polycystins may at least be partly responsible for the observed ciliary length phenotype of DLG1-deficient cells". I don't see how PC levels would cause changes in ciliary length. Has this been shown?

****Thank you for this comment. Shao et al. (2020) showed that "cilia are elongated in kidneys from patients with ADPKD and from both *Pkd1* and *Pkd2* knockout mice". Moreover, a study by David Clapham's group (Liu et al., eLife, 2018 PMID: 29443690) showed that conditional knockout of *Pkd1* or *Pkd2* in mouse kidney resulted in cyst formation and significant elongation of primary cilia in cells lining the cysts. The underlying mechanism is not clear (it could be secondary to altered signaling), but as we observed ciliary lengthening as well as reduced ciliary PC2 levels in *Dlg1* mutant kidney epithelial cells, we think it is relevant to mention that depletion of PC2/polycystins has been linked to cilia elongation. We have now also cited the manuscript by Liu et al. (2018) in the discussion as this paper had not been cited originally.**

Minor remarks

Line 53, please change "is" into "can be", as also mutations in other gene scan cause ADPKD.

****Done.**

As the authors use various cell lines in their analyses, it would be very helpful if they could also indicate in the figures themselves which cells were used. E.g. Fig 3A-D report on IMCD3 cells, whereas Fig3E-G are about mCCD cells. Please do so for all figures.

****Done.**

Fig 6J lacks statistical analysis. Please include this.

****Done.**

I found the conclusion sentence about the interactions between DLG1, SDCCAG3 and IFT20 a bit too strong. These are in vitro experiments with overexpressed proteins and modeling.

Please tune down this conclusion a bit and indicate that it is based on in vitro experiments and modeling.

****We have rephrased the relevant sentences as suggested.**

The results section ends with two very similar conclusion sentences. Please correct this.

****This has now been corrected.**

There were some English errors:

Line 147. Please add "of" between "amount" and "Dlg3".

****Done.**

Line 297, Please remove "in" at the end of the sentence.

****Done.**

Line 420, please change "we" into "were".

****Done.**

Referee #3

In this manuscript, Rezi et al. identify Dlg1 as a novel regulator of primary cilia. Dlg1 has known roles as a Scribble complex component in establishing basolateral polarity but prior studies have also suggested a link between cilia and Dlg1 on the basis of its localization to the centrosome in lung cancer cells and presence in the ciliary proteome of inner medullary collecting duct (IMCD3) cells. In their study, Rezi et al. assess the functional role of Dlg1 in kidney cilia, finding that conditional Dlg1 knockout in mouse kidney leads to structural defects at the organ level, as well as increased ciliary length and altered ciliary protein composition. Using cilia proximity labeling-based proteomic analysis in IMCD3 Dlg1 KO cells, the authors identified ciliary proteins that may be regulated by Dlg1, many of which are associated with proper cilia formation and maintenance as well as ciliary signaling. The authors focused on two hits from their proteomic screen: the retromer-associated protein SDCCAG3 and the IFT-B complex component IFT20, both of which have been previously implicated in trafficking of PKD2 to the cilium. Indeed, Dlg1 KO led to diminished SDCCAG3 localization to cilia in vivo and in IMCD3 cells, and to the ciliary base in mCCD cells. IFT20 protein levels were similarly reduced at the ciliary base in vivo and in mCCD cells. Furthermore, PC2 localization to the cilia was reduced in Dlg1 KO mCCD cells. The authors further note that a reported patient-derived Dlg1 mutant associated with congenital anomalies of the kidney and urinary tract (CAKUT) disorder failed to rescue IFT20 and SDCCAG3 localization in Dlg1 KO cells, hinting at a possible clinical connection between CAKUT and cilia. Lastly, coIPs and structural modeling suggested SDCCAG3 and IFT20 as interacting partners with each other and Dlg1 (albeit potentially indirect), strengthening the authors' Dlg1-dependent SDCCAG3 and IFT20 trafficking model. Taken together, these findings identify Dlg1 as a new potential ciliary regulator and offer some additional mechanistic insight (via IFT20 and SDCCAG3) and physiological relevance into Dlg1's ciliary role. Noted below are key questions that need to be addressed regarding some of the reported phenotypes and the potential connection to CAKUT. If the authors are able to clarify these issues, I believe their study would be of interest in the field and appropriate for EMBO Reports's audience.

****We thank the referee for an insightful and generally positive evaluation of our manuscript.**

Major points

1. What is the evidence for DLG1-T489R having any association with CAKUT? This mutant is not obviously listed in ref. 34 that is cited in the manuscript. This is an important point given that the use of this allele and the claimed relevance to CAKUT rest on it. Additionally, in the canonical DLG1 sequence (Uniprot Q12959), residue 489 is not a threonine. Clarification of the DLG1 sequence and which associated isoforms are expressed in human/mouse kidneys would be valuable.

****Unfortunately we had listed the wrong ref. 34. Instead of Nicolaou et al. 2015, ref. 34 should have been:**

Nicolaou, N. *et al.* Prioritization and burden analysis of rare variants in 208 candidate genes suggest they do not play a major role in CAKUT. *Kidney Int* 89, 476-486, doi:10.1038/ki.2015.319 (2016).

We apologize for this mistake which has now been corrected. In Supplementary Table 4 of Nicolaou et al. (2016) the DLG1 T489R variant was classified as a likely deleterious CAKUT variant due to the observation that a patient with this mutation exhibited duplex collecting system and multicystic kidney dysplasia. According to the Ensembl database, human *DLG1* (ENSG00000075711) has 71 transcripts; the specific *DLG1* transcript listed in Supplementary Table 4 of Nicolaou et al. (2016) is transcript ENST00000422288, which codes for DLG1 protein with Uniprot ID Q12959-5 (designated DLG1-210). This protein contains a Threonine residue at position 489. The rat homolog of human DLG1-210, which was used for mutagenesis and rescue experiments, is UniProt A0A8I6A5M7 and has the corresponding Threonine residue located at position 507, as indicated in Figure 6A. We have now updated the Figure 6A legend to clarify this.

2. It would be helpful for the authors to discuss any prior evidence that might suggest a link between cilia and CAKUT. The authors suggest that Dlg1-linked ciliary alterations may be responsible for CAKUT, but is it not also possible that Dlg1 separately impacts cilia and CAKUT?

****We agree with the Referee that it is currently unclear if Dlg1-linked ciliary alterations may be responsible for CAKUT and have toned down the relevant statements accordingly. It is certainly possible that Dlg1 separately impacts cilia and CAKUT, as stated by the Referee, and we have updated the relevant text in the Results section to reflect this point. Furthermore, at the end of our Discussion we specifically state that: “Nevertheless, if and how altered ciliary length and composition, as well as dysregulated metabolic, NFkB and TGFb signaling, contribute to the kidney defects observed in Dlg1 deficient mice and human CAKUT patients with DLG1 mutations awaits further investigation.” That said, several studies have indicated that mutations in ciliary genes can give rise to CAKUT, as described and discussed in a recent review by Gabriel et al. (Frontiers in Pediatrics, 2018). We now mention and cite this review in the Introduction.**

3. In the abstract, the authors report 'cystogenesis' in Dlg1 mutant kidney, but the data to support this seem scant. The Results report 'tubular dilations that appeared to be pre-cystic' but these are not rigorously or quantitatively characterized.

****As noted in the response to Referee 1, we would like to emphasize that the *Pax3Cre-Dlg1^{F/F}* mice do develop tubular dilations and cysts in the nephron of the cortical regions of the kidney. This cystic phenomenon was already rigorously described by the Miner lab upon ablation of *Dlg1* in the metanephric mesenchyme using *Six2-Cre* (Ahn et al, 2013), which is also expressed in the metanephric mesenchyme, and thus leads to deletion of *Dlg1* in the same nephron segments as *Pax3Cre*. We have included the relevant figure from the Ahn et al (2013) study below for the reviewer. Moreover, we have also included magnified images of the kidney cortex from *Pax3Cre-Dlg1^{F/F}* mice to highlight these cystic dilations (new Figure 1A). We have added these details to the Results section.**

4. While the authors claim that *Dlg1* regulates ciliary length indirectly as a principally plasma membrane-localized protein, they also note that over-expressed GFP-*Dlg1* localized to the cilium base in RPE1 cells. Did the authors analyze the localization of overexpressed *Dlg1*-mCh in mCCD and IMCD3 cells (and not just RPE1 cells)? Conversely, what is the localization of endogenous *Dlg1* in RPE1 cells?

****Please see responses to similar comments from Referee #1 and #2 above.**

5. The authors claim that SDCCAG3 is a retromer subunit, but is this true? Evidence seems to indicate it can be identified via IP/MS as a Vps35-associated protein (ref. 41), but not necessarily as a bona fide subunit of retromer. In the absence of further evidence that SDCCAG3 is part of a bona fide retromer complex, it would seem more appropriate to call it a retromer-associated protein or binding partner.

****Thank you for pointing this out, we have updated the relevant sentences as suggested.**

6. It is a bit odd that there is a cell-type-specific effect of *DLG1* KO on IFT20 localization - IFT20 is reduced in mCCD *Dlg1* KO cells but not in mutant IMCD3 cells, even though mutant IMCD3 cells were used in the proximity labeling experiments that first identified an effect of *Dlg1* KO on ciliary IFT20 levels. It would be helpful if the authors provided more of an explanation for this apparent discrepancy.

****Since IFT20 has been shown to be associated with the Golgi and due to the close proximity of the Golgi to the primary cilium, the conventional fixation methods make it difficult to quantify the ciliary levels of IFT20. To overcome this, we used a different fixation method described by Hua & Ferland 2017, which optimizes the ciliary staining of IFT20 in the cells, along with disrupting the Golgi staining of IFT20. However, with this approach in IMCD3 cells, we encountered a higher background, which most probably impacted our results. Although we did not observe a significant alteration, we did notice a tendency toward lower levels of IFT20 at the ciliary base in IMCD3 cells compared to WT. This is now clarified in the Results section.**

7. In Fig. 5 (and possibly 3E), the contrast of the microscopy images appears to be saturated, precluding a direct comparison of the intensity of SDCCAG3 and PC2 levels across the different experimental conditions. Despite this, in Fig 5C and 5E, the PC2 and SDCCAG3 intensity and ciliary localization do not look to be restored to wildtype-like status upon expression of mCh-DLG1, even the the quantifications do indicate full rescue. Can the authors clarify this issue? How exactly were signals quantified?

****The quantitative analysis was done on Z-stacks whereas the IFM figures were generated by deconvolution of the Z-stacks, which may have caused the images to appear saturated. We have now repeated this analysis and Figure 5C and 5E have been replaced with better quality images.**

8. In Fig. 6F, the basal body levels of IFT20 appear to be significantly lower in cells expressing the T507R mutant than in DLG1 KO cells. Can the authors comment on this or suggest an explanation?

****We have now included data for the WT DLG1 rescue construct in the comparative quantitative IFM analysis of IFT20 and SDCCAG3 ciliary levels of our mCCD cell lines (new Figure 6F and G). This new comparative analysis shows that the difference in relative ciliary base levels of IFT20 of the *Dlg1*^{-/-} and the DLG1^{T507R} rescue line is very small. While we do not have an explanation for this small difference, we suspect that the DLG1^{T507R} variant could have a dominant negative effect, e.g. perhaps by somehow affecting the function of DLG4, which is also expressed in these cells (Figure S1). Given the modest magnitude of the difference, we decided not to comment on it in the text.**

9. What is the effect of the DLG1-T507R on ciliary PC2 levels?

****Thank you for this question. We have now performed additional quantitative IFM analysis of our different mCCD cells lines, including the DLG1^{T507R} rescue line, grown on Transwell filters and stained with PC2 antibody. The results of this analysis, which are included in the new Figure S6, indicate that the DLG1^{T507R} variant can restore ciliary PC2 levels to some degree, but not as efficiently as the WT DLG1 construct, i.e. a partial rescue. We have mentioned this in the Results section: *"We also tested the impact of the DLG1^{T507R} variant on the ciliary levels of PC2 by IFM analysis of transwell filter-grown mCCD cells. Our findings indicated that the DLG1^{T507R} variant can partially restore the ciliary levels of PC2 to normal in the *Dlg1*^{-/-} background although not to the same extent as WT DLG1 (Figure S6). This suggests that this specific point mutation in DLG1 has a more severe impact on ciliary targeting of SDCCAG3 and IFT20 than PC2, implying that DLG1 promotes ciliary PC2 trafficking not only via SDCCAG3 and IFT20."***

10. Alpha Fold reported identification of a "high confidence interaction between IFT20 and SDCCAG3" (lines 308-309). What is the evidence that the interaction is "high confidence"?

**** The high confidence for the SDCCAG3-IFT20 interaction is evidenced by the predicted alignment error (PAE) score shown in Figure S6B. The boxed area of the PAE plot in Figure S7B (former Figure S6B) demonstrates low error along a diagonal consistent with SDCCAG3 and IFT20 forming an antiparallel coiled-coil structure. The PAE scores are <10Å along the hetero-dimeric coiled coil (and <5Å for most of the length), consistent with a high-confidence structural prediction of the complex structure. We have amended the figure legend for Figure S7B to provide a better explanation of this.**

11. The wording in the abstract "using biochemical approaches and AlphaFold modelling we show that DLG1 associates physically with SDCCAG3 and IFT20" is perhaps confusing/misleading as it arguably implies that Alpha Fold predicted interactions between Dlg1 and IFT20, or between Dlg1 and SDCCAG3, which apparently was not the case.

****We apologize for the confusion and have now updated the abstract to clarify this point.**

Minor points

1. Why is it important that Pax3Cre is active in urogenital mesenchyme (line 135)? The relevance of this to the present findings is unclear.

****We simply point this out because the loss of Dlg1 in the urogenital mesenchyme (via Pax3Cre) is the cause of the observed hydronephrosis in the kidneys of our mice (Figure 1A).**

2. In line 172 of the manuscript, the text refers to using Arl13b as the ciliary marker in Fig. S2B, but AcTub is actually shown. Either the figure or the text should be corrected.

****Thank you for noticing this mistake. The ciliary marker used is AcTub and this has now been corrected in the text.**

3. Not only are cilia longer in Dlg1 knockouts versus WT, but the intensity of the ciliary marker AcTub appears to be greatly increased in vivo (see Fig. 1B, 4A). Is this in fact the case or observed in cell lines?

****The increase in the acetylated tubulin staining is due to the fact that the cilia are longer, and thus there is "more" overall signal for acetylated tubulin. We have now quantified the levels of acetylated tubulin in control and Dlg1 KO kidneys (new Figure 1D). As expected, there is an increase in total acetylated tubulin intensity per cilium, consistent with the increase in ciliary length (Figure 1C). This information has been added to the Results section.**

4. Could the authors provide a comment on the shape of the volcano plot and why multiple hits having apparently maximal magnitudes of enrichment/depletion (albeit different p-values)? Additionally, why is Dlg1 not more strongly depleted?

**** We apologize for the lack of clarity. There are multiple points on either side of the x-axis that have the same log2 fold change values as they have median intensity values equal to 0 for either the WT or the KO. As a result, the ratio is infinite and we would not be able to plot these values. Thus, we introduced a pseudo ratio value for such genes that is 2 units above or below the maximum and minimum ratio value across all genes**

respectively. Hence all these points have a single x-axis value. However, each one of them also have a t-test comparison associated with them that give rise to a different p-value owing to the differing intensity values for each gene, thereby having differing y-axis values. We have now added some text to the legend of Figure 2 explaining this.

We understand that in the context of the whole volcano plot, it may seem that the *Dlg1* is not as depleted as many other genes. However, the depletion is more than 10-fold (\log_2 fold change of -3.43). On further inspection, we in fact noticed that one of the replicates for the DLG1-KO had a high intensity value (see plot below) thereby driving the median a little bit towards the higher side. This also affected the outcome of the t-test. Thus overall, *Dlg1* is heavily depleted in 5 out of the 6 replicate samples and the ratio value in the volcano plot is influenced by the outlier sample.

Figure: Density histogram showing the LFQ intensity values for the *Dlg1* gene in the six replicate samples of the WT and DLG1 KO. The colored lines show the median LFQ intensity values for the respective strains. The DLG1 KO had 5 replicate samples with intensity values > 10-fold lower than the WT samples.

5. In line 210 of the manuscript, reference 15 appears to show SDCCAG3 localization to the basal body/centrioles rather than to the primary cilia.

****Thank you for pointing this out, we have now corrected the sentence accordingly.**

6. In Fig. 3B and 3F, significance tests should be added comparing SDCCAG3 localization between WT v. *Dlg1*^{-/-} mCh-Dlg1.

****Done.**

7. What is quantified in Fig. 4B? Is the fluorescence intensity the average across many cells/cilia or from different samples/kidneys analyzed?

****For this figure, we quantified the total levels of SDCCAG3 and IFT20 in cilia of control (N=3) and *Dlg1* KO (N=4) mice. The levels from control mice were set to 1, and the ciliary levels from mutant mice were compared to that (i.e., relative fluorescence intensity). Data shown are the average values from each mouse. This information is added to the Methods section and the legend of Figure 4B.**

8. Asterisks indicating the ciliary base are missing from Fig. 5C and 5E.

****Asterisks have now been included in the new Figures 5C and 5E.**

9. For Fig. S5, what would happen if the ALVX-144 and ReTAT-N-dimer peptides are combined? Would there be a more drastic drop in PC2 levels at the cilia/centrosome? Additionally, it would be informative to include significance values comparing 0uM and 25uM ReTat-N-dimer in Fig. S5B (top panel). Could the authors comment on why treating cells with a 25uM concentration of ReTat-N-dimer results in a less significant result than 5uM in Fig. S5B (bottom panel)?

**** ALVX-144 and ReTAT-N-dimer are highly similar compounds that target the same region of DLG1 but with different binding affinity values (Bach et al. 2012). Therefore, combining the two compounds would not add much new information. In the case of the 25uM ReTat-N-dimer treatment, where we did not observe significant changes in the PC2 levels compared to the untreated cells, we speculate that cells may be compensating somehow as we increase the concentration of the inhibitor. To avoid confusion, we removed the data on the 25uM ReTat-N-dimer treatment from Figure S5B.**

10. For Fig. 5D, the P values between comparing PC2 levels in WT v. *Dlg1*^{-/-} mCh-Dlg1 seem to indicate highly significant differences. Are these correct, which would suggest incomplete rescue by the transgene?

****Thank you for noticing this. We have repeated the quantitative and statistical analysis and now find that there is no significant difference between the WT and *Dlg1*^{-/-} mCh-Dlg1 rescue line. Figure 5D has been updated accordingly.**

11. Fig 5C. The green channel appears to be missing from top left sub-panel.

****Sorry for the oversight. There was a problem with deconvolution of the Z-stacks used for generating these figure panels. This problem has now been fixed and Figure 5C and 5E have been replaced with better quality images.**

12. Are the regions of IFT20/SDCCAG3 predicted by Alpha Fold to mediate their association also functionally required for a co-IP between these two proteins?

****We agree that it would be interesting to map the actual interaction site(s) in SDCCAG3 and IFT20, but due to limited time available for revision we have prioritized addressing other experiments requested by the Referees, including additional IFM analyses and fluorescence intensity quantifications. Moreover, we note that the EMBO Reports homepage specifically states that "When considering submissions, the journal focuses on novelty and the physiological and/or functional significance of a finding, rather than the level of mechanistic detail reported."**

Dear Lotte,

Thank you for the submission of your revised manuscript to EMBO reports. I had already sent you the referee reports (copied again below) and have now finalized all editorial checks. Below I list a number of points that we need from the editorial side.

- Please reduce the number of keywords to 5.
- Data availability: Is it possible to deposit the mass spectrometry data in a relevant public repository? In case, please also update the information in the Author Checklist, which will be published alongside your manuscript (as part of the review file).
- The script developed by DBI-INFRA on github. If the script has already been deposited on github, and I recommend so because all data should be publicly available, then please update the description in the Materials and Methods section and please provide a link to the script on github in the Data availability section.
- Regarding the mouse kidney paraffin sections: did you obtain these yourself from the mice or did you obtain the already sectioned and paraffin-embedded samples from another lab? The reason why I am asking is that in the former case you need to report on housing and husbandry conditions as well as ethical approval of mouse experiments.
- Please note that as per our editorial policy, all data mentioned must be included in the manuscript. In this respect, we noticed that you based one of your statements on 'data not shown' on page 8 of the manuscript. Please either include the data, e.g., in the Appendix, or remove the statement from the manuscript.
- Reagents table (named Table 1) should be removed from the manuscript and uploaded as a separate Word file called Reagents and Tools table. Please use the Reagents and Tools table template that you can download from our Guide to Authors.
- Table S1 is a Dataset - it needs to be renamed and uploaded as Dataset EV1; the callout in the manuscript needs to be updated too. Please add a legend in a separate tab in the .xls file.
- Supplementary figures: They can remain within the single PDF but please rename them to Appendix Figure S1, etc. The PDF file is called 'Appendix'. We also need a title page with a table of content and page numbers. Please also update the callouts in the manuscript text. Please note that you can promote up to 5 figures to the Expanded View content (Figure EV#). If you choose that option then each EV figure needs to be uploaded separately and their legends need to be provided in the manuscript file. The other 3 figures would remain part of the Appendix in this case.
- Figure callouts:
 - a) Figure 4 is missing callouts for the individual panels A and B;
 - b) All references to Suppl. Figures and Table need to be correctly updated to e.g. Appendix Figure S#, or Figure EV# and Dataset EV1.
- All source data of one figure need to be zipped up into one folder and in the end, source data should be uploaded as one folder per figure.
- Summary should be renamed to Abstract.
- Please add a 'Disclosure and competing interests statement'. For more information see <https://www.embopress.org/page/journal/14693178/authorguide#conflictsofinterest>
- The manuscript sections should be in the following order: Title page - Abstract & Keywords - Introduction - Results - Discussion - Materials & Methods - Data Availability - Acknowledgments - Disclosure Statement & Competing Interests - References - Figure Legends - Tables with legends - Expanded View Figure Legends.
- The references need to be alphabetical, et al should be used after 10 author names, year should be in brackets; in-text citations have square brackets.
- Our production/data editors have asked you to clarify several points in the figure legends (see below). Please incorporate these changes in the manuscript and return the revised file with tracked changes with your final manuscript submission.
 - a) Please note that a separate 'Data Information' section is required in the legends of figures 1c-d, g-h.
 - b) Please note that the legends for figures 3b-f is not provided in the sequential manner (legend for figure 3e is provided before legends of figures 3b-d, and legend for figure 3f is provided before legends of figures 3c-d). This needs to be rectified.

c) Please note that the figure panel 4b is not labelled in the figure, also the figure panel 4c is mislabeled as 4b in the figure. This needs to be rectified.

d) Please note that the legends for figures 5d-e is not provided in the sequential manner (legend for figure 5e is provided before legend of figure 5d). This needs to be rectified.

e) Please note that the legends for figures 6i-j is not provided in the sequential manner (legend for figure 6j is provided before legend of figure 6i). This needs to be rectified.

f) Please note that the legends for supplementary figures 3b-c is not provided in the sequential manner (legend for figure 3c is provided before legend of figure 3b). This needs to be rectified.

g) Please note that the legends for supplementary figures 4b-e is not provided in the sequential manner (legend for figure 4d is provided before legends of figures 4b-c, and legend for figure 4e is provided before legend of figure 4c). This needs to be rectified.

h) Please note that the legends for supplementary figures 5b-e is not provided in the sequential manner (legend for figure 5c is provided before legend of figure 5b). This needs to be rectified.

i) Please define the annotated p values ****/*/* in the legend of figure 1g; 3b, f; 4c; 5d, f; 6c, f-g, k; as appropriate.

j) Please indicate the statistical test used for data analysis in the legends of figures 2a; 4c; 6c, i, k.

k) Please note that the box plot needs to be defined in terms of minima, maxima, centre, bounds of box and whiskers, and percentile in the legend of figure 4c.

l) Although 'n' is provided, please describe the nature of entity for 'n' in the legends of figures 3b, f; 6c, f-g, i, k, supplementary figures 1a-c, e-h; 4c, f; 5b, d.

m) Please note that the error bars are not defined in the legends of figures 1g-h; 3b, f; 4c; 5d, f; 6c, f-g, i, k.

n) Please note that scale bar and its definition are missing for supplementary figures 1d-e.

o) Please note that the asterisk is not defined in the legend of figure 5c, e. This needs to be rectified.

- Appendix figure legends: in general, please define the size of the scale bars in the figure legend.

- Please define the nature of the replicates (technical, biological/independent) for Fig S1A-C, S2E-H, S4C, F, S5B, D.

- Please add a scale bar in panel 'E'.

- Please describe all findings in the abstract in present tense.

- Finally, EMBO Reports papers are accompanied online by A) a short (1-2 sentences) summary of the findings and their significance, B) 2-3 bullet points highlighting key results and C) a synopsis image that is 550x300-600 pixels large (width x height) in PNG for JPG format. You can either show a model or key data in the synopsis image. Please note that the size is rather small and that text needs to be readable at the final size. Please send us this information along with the revised manuscript.

- On a different note, I would like to alert you that EMBO Press offers a new format for a video-synopsis of work published with us, which essentially is a short, author-generated film explaining the core findings in hand drawings, and, as we believe, can be very useful to increase visibility of the work. This has proven to offer a nice opportunity for exposure i.p. for the first author(s) of the study. Please see the following link for representative examples and their integration into the article web page:

<https://www.embopress.org/doi/full/10.15252/embj.2019103932>

With kind regards,

Martina

Referee #1:

All my concerns have been addressed by the authors. I understand that additional mice are no longer available and, also in the light of 3R, should not be generated again for this particular question.

Referee #2:

I'm happy with the changes and explanations made by the authors and have no further comments.

Referee #3:

The authors have now satisfactorily addressed the concerns raised previously, and I feel the revised manuscript is suitable for publication in EMBO Reports. My only minor comment is that despite the new explanation by the authors, the ciliary acetylated tubulin intensity in Dlg1 KO cells seems greater per unit area and not just higher due to cilia elongation. Can the authors quantify the acetylated tubulin signal per unit area or pixel to assess this?

Copenhagen, May 08, 2024

Dear Dr. Rembold,

Thank you for the EMBOR-2023-58624V2 Decision Letter. We have now carried out the revisions and edits requested, as indicated in bold text font below.

Our responses to specific comments by editor and referees

- Please reduce the number of keywords to 5. **Done**

- Data availability: Is it possible to deposit the mass spectrometry data in a relevant public repository? In case, please also update the information in the Author Checklist, which will be published alongside your manuscript (as part of the review file). **The mass spectrometry proteomics data have been deposited to the ProteomeXchange Consortium via the PRIDE (Perez-Riverol *et al*, 2022) partner repository with the dataset identifier PXD051912. We have included this information in the Data availability section of our manuscript, as well as in the Author Checklist.**

- The script developed by DBI-INFRA on github. If the script has already been deposited on github, and I recommend so because all data should be publicly available, then please update the description in the Materials and Methods section and please provide a link to the script on github in the Data availability section. **The script has been deposited on github and we have added the necessary information in the Materials and Methods and data Availability Section.**

- Regarding the mouse kidney paraffin sections: did you obtain these yourself from the mice or did you obtain the already sectioned and paraffin-embedded samples from another lab? The reason why I am asking is that in the former case you need to report on housing and husbandry conditions as well as ethical approval of mouse experiments. **We have added the necessary information in the Materials and Methods on page 25.**

- Please note that as per our editorial policy, all data mentioned must be included in the manuscript. In this respect, we noticed that you based one of your statements on 'data not shown' on page 8 of the manuscript. Please either include the data, e.g., in the Appendix, or remove the statement from the manuscript. **We have removed the statement from page 8.**

- Reagents table (named Table 1) should be removed from the manuscript and uploaded as a separate Word file called Reagents and Tools table. Please use the Reagents and

Tools table template that you can download from our Guide to Authors. **Done.**

- Table S1 is a Dataset - it needs to be renamed and uploaded as Dataset EV1; the callout in the manuscript needs to be updated too. Please add a legend in a separate tab in the .xls file. **Done.**

- Supplementary figures: They can remain within the single PDF but please rename them to Appendix Figure S1, etc. The PDF file is called 'Appendix'. We also need a title page with a table of content and page numbers. Please also update the callouts in the manuscript text. Please note that you can promote up to 5 figures to the Expanded View content (Figure EV#). If you choose that option then each EV figure needs to be uploaded separately and their legends need to be provided in the manuscript file. The other 3 figures would remain part of the Appendix in this case. **We have combined all our Supplementary figure files (Figure S1-S8) and accompanying figure legend in a pdf file called "Appendix". This file now includes a title page with a table of content and page numbers. Relevant callouts in the manuscript text have been updated.**

- Figure callouts:

a) Figure 4 is missing callouts for the individual panels A and B. **Figure 4 panels were mislabeled (panel C label missing), this has now been corrected and relevant callouts have been updated in the manuscript text.**

b) All references to Suppl. Figures and Table need to be correctly updated to e.g. Appendix Figure S#, or Figure EV# and Dataset EV1. **Done.**

- All source data of one figure need to be zipped up into one folder and in the end, source data should be uploaded as one folder per figure. **Done.**

- Summary should be renamed to Abstract. **Done.**

- Please add a 'Disclosure and competing interests statement'. For more information see <https://www.embopress.org/page/journal/14693178/authorguide#conflictsofinterest> **Done.**

- The manuscript sections should be in the following order: Title page - Abstract & Keywords - Introduction - Results - Discussion - Materials & Methods - Data Availability - Acknowledgments - Disclosure Statement & Competing Interests - References - Figure Legends - Tables with legends - Expanded View Figure Legends. **Done.**

- The references need to be alphabetical, et al should be used after 10 author names, year should be in brackets; in-text citations have square brackets. **Done.**

- Our production/data editors have asked you to clarify several points in the figure legends (see below). Please incorporate these changes in the manuscript and return the

revised file with tracked changes with your final manuscript submission.

a) Please note that a separate 'Data Information' section is required in the legends of figures 1c-d, g-h. **Relevant data information has now been included in the Figure 1 legend.**

b) Please note that the legends for figures 3b-f is not provided in the sequential manner (legend for figure 3e is provided before legends of figures 3b-d, and legend for figure 3f is provided before legends of figures 3c-d). This needs to be rectified. **This has been corrected.**

c) Please note that the figure panel 4b is not labelled in the figure, also the figure panel 4c is mislabeled as 4b in the figure. This needs to be rectified. **Figure 4 panels were mislabeled (panel C label missing), this has now been corrected and relevant callouts have been updated in the manuscript text.**

d) Please note that the legends for figures 5d-e is not provided in the sequential manner (legend for figure 5e is provided before legend of figure 5d). This needs to be rectified. **Done.**

e) Please note that the legends for figures 6i-j is not provided in the sequential manner (legend for figure 6j is provided before legend of figure 6i). This needs to be rectified. **Done.**

f) Please note that the legends for supplementary figures 3b-c is not provided in the sequential manner (legend for figure 3c is provided before legend of figure 3b). This needs to be rectified. **Done.**

g) Please note that the legends for supplementary figures 4b-e is not provided in the sequential manner (legend for figure 4d is provided before legends of figures 4b-c, and legend for figure 4e is provided before legend of figure 4c). This needs to be rectified. **Done.**

h) Please note that the legends for supplementary figures 5b-e is not provided in the sequential manner (legend for figure 5c is provided before legend of figure 5b). This needs to be rectified. **Done.**

i) Please define the annotated p values ****/**/* in the legend of figure 1g; 3b, f; 4c; 5d, f; 6c, f-g, k; as appropriate. **Done.**

j) Please indicate the statistical test used for data analysis in the legends of figures 2a; 4c; 6c, i, k. **Done.**

k) Please note that the box plot needs to be defined in terms of minima, maxima, centre, bounds of box and whiskers, and percentile in the legend of figure 4c. **Done.**

l) Although 'n' is provided, please describe the nature of entity for 'n' in the legends of figures 3b, f; 6c, f-g, i, k, supplementary figures 1a-c, e-h; 4c, f; 5b, d. **Done.**

m) Please note that the error bars are not defined in the legends of figures 1g-h; 3b, f; 4c; 5d, f; 6c, f-g, i, k. **This has now been corrected.**

n) Please note that scale bar and its definition are missing for supplementary figures 1d-e. **This has now been corrected.**

o) Please note that the asterisk is not defined in the legend of figure 5c, e. This needs to be rectified. **Done.**

- Appendix figure legends: in general, please define the size of the scale bars in the figure legend. **We have defined the size of scalebars in the images, we hope this is acceptable?**

- Please define the nature of the replicates (technical, biological/independent) for Fig S1A-C, S2E-H, S4C, F, S5B, D. **Done.**

- Please add a scale bar in panel 'E'. **Done.**

- Please describe all findings in the abstract in present tense. **Done.**

- Finally, EMBO Reports papers are accompanied online by A) a short (1-2 sentences) summary of the findings and their significance, B) 2-3 bullet points highlighting key results and C) a synopsis image that is 550x300-600 pixels large (width x height) in PNG for JPG format. You can either show a model or key data in the synopsis image. Please note that the size is rather small and that text needs to be readable at the final size. Please send us this information along with the revised manuscript.

We have uploaded the requested files.

Referee #3:

The authors have now satisfactorily addressed the concerns raised previously, and I feel the revised manuscript is suitable for publication in EMBO Reports. My only minor comment is that despite the new explanation by the authors, the ciliary acetylated tubulin intensity in Dlg1 KO cells seems greater per unit area and not just higher due to cilia elongation. Can the authors quantify the acetylated tubulin signal per unit area or pixel to assess this?

We have quantified the intensity of acetylated tubulin per unit length of cilia. Indeed, there is a slight but significant increase in acetylation per unit length cilium in a fraction of the cell population in the Dlg1 KO mice. This suggests a potential defect in posttranslational modification of cilia in the absence of Dlg1, which may be direct or indirect. The new data is presented as new Appendix Figure S1.

Dear Lotte,

Thank you for submitting your revised manuscript. I have now looked at everything and all is fine. Therefore, I am very pleased to accept your manuscript for publication in EMBO Reports.

Congratulations on a nice work!

Kind regards,

Deniz

--

Deniz Senyilmaz Tiebe, PhD

Editor

EMBO Reports
